# Debiased and Denoised Entity Recognition from Distant Supervision

**Haobo Wang**[1*], **Yiwen Dong**[1*], **Ruixuan Xiao**[1], **Fei Huang**[2], **Gang Chen**[1], **Junbo Zhao**[1†]

[1]Zhejiang University, Hangzhou, China
[2]Alibaba Group, Hangzhou, China
{wanghaobo, dyw424, xiaoruixuan, cg, j.zhao}@zju.edu.cn, feirhuang@gmail.com

## Abstract

While distant supervision has been extensively explored and exploited in NLP tasks like named entity recognition, a major obstacle stems from the inevitable noisy distant labels tagged unsupervisedly. A few past works approach this problem by adopting a self-training framework with a sample-selection mechanism. In this work, we innovatively identify two types of biases that were omitted by prior work, and these biases lead to inferior performance of the distant-supervised NER setup. First, we characterize the noise concealed in the distant labels as highly structural rather than fully randomized. Second, the self-training framework would ubiquitously introduce an inherent bias that causes erroneous behavior in both sample selection and eventually prediction. To cope with these problems, we propose a novel self-training framework, dubbed DesERT. This framework augments the conventional NER predicative pathway to a dual form that effectively adapts the sample-selection process to conform to its innate distributional-bias structure. The other crucial component of DesERT composes a debiased module aiming to enhance the token representations, hence the quality of the pseudo-labels. Extensive experiments are conducted to validate the DesERT. The results show that our framework establishes a new state-of-art performance, it achieves a **+2.22%** average F1 score improvement on five standardized benchmarking datasets. Lastly, DesERT demonstrates its effectiveness under a new DSNER benchmark where *additional distant supervision* comes from the ChatGPT model.

## 1   Introduction

As an iconic and pivotal task, named entity recognition (NER) [1] has gained wide attention from the natural language processing community. While there have been several successes towards solving this task [2, 3, 4, 5], most of these solutions demand decent datasets for training that enjoys both good quality and large quantity. When it comes to specialized industrial deployment, this leads us to expensive data annotation and curation costs which may hinder the development of NER.

Distant supervision [6, 7, 8] is a promising direction to alleviate this problem. Indeed, there have been a number of attempts to combine the additional labels obtained from distant supervision with the original unlabeled data, and then establish a selection-based self-training framework [9] to handle noise among distant labels. The core of these methods is to sample reliable clean tokens and cope with potentially noisy tokens involved by the imperfect pseudo-label prediction.

On one hand, we characterize that the label distribution for distant supervision is rather skewed. Notably, as shown in Figure 1 (a), we found that true non-entity tokens are far more than the remaining

---

[*]Equal contribution.
[†]Corresponding author.

37th Conference on Neural Information Processing Systems (NeurIPS 2023).

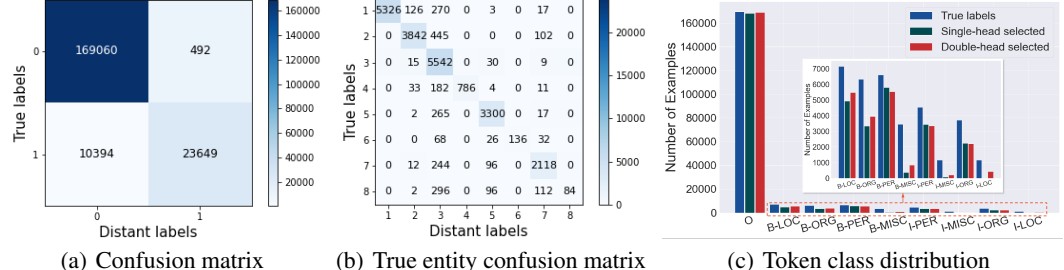

(a) Confusion matrix      (b) True entity confusion matrix      (c) Token class distribution

Figure 1: (a) Confusion matrix of true labels and distant labels on the whole CoNLL03 dataset. Class index 0 denotes non-entity and 1 denotes all entity types, more than 169.1k non-entity tokens are correctly labeled and dominate true entity tokens. (b) The confusion matrix displays noise among true entity-type labels. (c) The real token class distribution on the CoNLL03 dataset and tokens selected at the fifth epoch by a basic self-training framework with a single-head/double-head pathway. It can be shown that double-head selects more tokens than single-head on the minority entity classes such as MISC and LOC.

tokens, quantity-wise. Meanwhile, there exists a severe class imbalance in true entity tokens with fine-grained noise in Figure 1 (b). Ignoring these phenomena by prior sampling-based methods is potentially performance-limited. The reason is that the dominant clean non-entity tokens can be easily distinguished and excessively picked up, while clean entity tokens are largely overlooked, especially on label-deficient entities like MISC and LOC; see Figure 1 (c). Whereas, the clean entity tokens are of real interest since NER attempts to distinguish the fine-grained token types. Unfortunately, there has not been any prior work that discusses or addresses this property specifically.

On the other hand, in spite of that self-training scheme applied with distant supervision is proven applicable [10, 9, 11], it could still be unreliable mostly due to the deviation between the pseudo-labels and unobservable true labels. In other words, the self-training scheme is inherently prone to assign erroneous pseudo-labels, which can be aggravated due to label noise and finally cause error accumulation and conversely pollutes the selection. We postulate that both biases ought to be properly resolved in order to fully unlock the utility boost of the distant supervision technique for NER.

To this end, we propose DesERT, a holistic framework that resolves the aforementioned problems and achieves state-of-the-art DSNER performance. The main idea behind the DesERT is to decouple the initial task/data into entity vs. non-entity and fine-grained entity type sub-parts, then respectively to split again by selecting and splitting out clean- and noisy-token sets which are processed by different learning paradigms, forming a task/data taxonomy. We further introduce debiased self-training on noisy sets to reduce erroneous pseudo-labels. Empirical findings successfully justify the validity of our approach, where on multiple standardized NER benchmarks we establish a new set of state-of-the-art results. For example, DesERT achieves state-of-art performance, especially an improvement of **3.26%** on the CoNLL03 dataset and **4.26%** on the Webpage dataset. Finally, we establish a new benchmark for DSNER where the supervision signals are generated via the large language model ChatGPT and DesERT still holds its superiority.

## 2 Related Works

**Distantly-Supervised NER.** There are two typical forms of noise in the labeling of distantly-supervised NER (DSNER) tasks: false negative tokens wrongly tagged as non-entities and inaccurate entity types. A series of works mainly focus on processing false negative tokens. A popular line of works adopts PU Learning [12, 13] techniques to resolve DSNER [14, 15], which sees entity tokens as Positive (P) data and non-entity tokens as Unlabeled (U) data, yet ignore the noise in fine-grained entities. Other methods consider both types of noise, usually inspired by traditional NLL approaches. One of them is based on clean token selection. Lots of works adopt the model's prediction consistency or confidence as the selection criterion [16, 17, 10, 18, 9, 11, 19, 20]. Besides, Huang et al. [21] propose a novel Logit Maximum Difference (LMD) score. HGL [22] introduces hypergeometric distribution to estimate noisy sample size. Additionally, some studies [23, 24] also employ reinforcement learning agents to filter noisy samples. A recent work [25] studies a specific DSNER setup that collects a large set of weakly labeled data and a small set of clean training data. The

most related work to ours is [26] which also investigates the bias issues in DSNER, but it concentrates on structural causal biases caused by the dictionaries. In contrast, our work focuses on unexpected selection and confirmation biases in popular *selection and self-training* DSNER frameworks.

**Self-training.** Self-training [27, 28, 29] is a popular semi-supervised learning technique that trains on labeled data but assigns pseudo-labels to unlabeled data for further training [30]. This is also adopted by existing DSNER approaches. For example, BOND [10] uses pseudo-labels from the teacher model to train the student model. SCDL [9] jointly trains two teacher-student networks to mutually provide pseudo-labels. RoSTER [11] ensembles different models' predictions as pseudo-labels. However, some research has pointed out that self-training exists an inherent confirmation bias to assign erroneous pseudo-labels [31, 32]. Due to the existence of label noise, the DSNER task can face an amplified self-training bias, which, however, has never been touched on in prior studies.

## 3 Notation and Preliminary

**Named Entity Recognition.** Assume we receive a NER training dataset $\mathcal{D} = \{(\boldsymbol{x}_m, \boldsymbol{y}_m)\}_{m=1}^M$ with $M$ tuples, where each tuple consists of a sentence $\boldsymbol{x} = [x_1, x_2, ..., x_n]$ with $n$ tokens and a label sequence $\boldsymbol{y} = [y_1, y_2, ..., y_n]$. Here each entity label $y_i$ indicates a token $x_i$ is a non-entity or belongs to a specific entity type. $y_i \in \mathcal{T} = \{0, 1, ..., K\}$, where $1 \sim K$ denote entity types and $0$ denotes non-entity. Our goal is to train a NER model $f$ parameterized by $\theta$ to be able to receive a testing sentence and predict the entity labels for its tokens. For fully supervised NER, this goal can be achieved by minimizing the empirical standard cross-entropy loss as follows,

$$\mathcal{L}_{ce}(\boldsymbol{y}, f(\boldsymbol{x}; \theta)) = -\frac{1}{n} \sum_{i=1}^n \log \text{softmax}(f_{i,y_i}(\boldsymbol{x}; \theta))$$

where $f_{i,y_i}(\boldsymbol{x}; \theta)$ is the model's predicted probability of $i$-th token $x_i$ belonging to class $y_i$ in sentence $\boldsymbol{x}$. In practice, the NER model $f$ is typically composed of a pre-trained language model encoder $\phi$ for extracting token representation and a classification head $h$, i.e., $f(\boldsymbol{x}; \theta) = h \circ \phi(\boldsymbol{x})$.

**Distantly-Supervised NER.** This paper considers the distantly-supervised NER problem, which effectively reduces labeling costs. Accordingly, the true label sequences $\{\boldsymbol{y}_m\}_{m=1}^M$ are replaced by their distantly-annotated counterparts $\{\tilde{\boldsymbol{y}}_m\}_{m=1}^M$. Despite the promise, the distant labels inevitably contain large portions of label noise due to the limited coverage of entity mentions in the knowledge base and the labeling ambiguity. To cope with this problem, we follow previous works [10, 9] and introduce the *selection and self-training* framework to mitigate the label noise. At each epoch, we first perform token selection based on the specific criterion that separates the tokens in each sentence $\boldsymbol{x}$ into a set of potential clean tokens $\boldsymbol{x}^l$ and noisy tokens $\boldsymbol{x}^u$, and thus, a robust NER model can be trained on those clean tokens. To further exploit the remaining noisy tokens, we regard them as unlabeled tokens and perform semi-supervised training,

$$\mathcal{L}_{cls} = \mathcal{L}_L(\tilde{\boldsymbol{y}}^l, f(\boldsymbol{x}^l; \theta)) + \mathcal{L}_U(\hat{\boldsymbol{y}}^u, f(\boldsymbol{x}^u; \theta))$$

where $\mathcal{L}_L$ and $\mathcal{L}_U$ are classification losses on the two sets. $\hat{\boldsymbol{y}}^u$ is a pseudo-label generated via previous models. In the sequel, we will specify our selection criterion and self-training algorithms in detail.

## 4 Proposed Method: DesERT

In theory, distant supervision can provide an arbitrarily large dataset for NER with its scale upper-bounded by all available sources. In this task of distantly-supervised NER, we denote the initial set extracted by distant supervision by $\mathcal{D}$. Of course, directly using $\mathcal{D}$ to train a NER model by supervised learning is problematic, due to the noise, uncertainty, or mismatches hidden in $\mathcal{D}$. In this section, we provide a detailed description of our framework. The resulting architecture would effectively decompose $\mathcal{D}$ into a task-data taxonomy, with each component from this taxonomy manifesting an individual training paradigm. This decomposition is stemmed from the two bias problems we mentioned previously. The pseudo-code of DesERT is summarized in Appendix D.

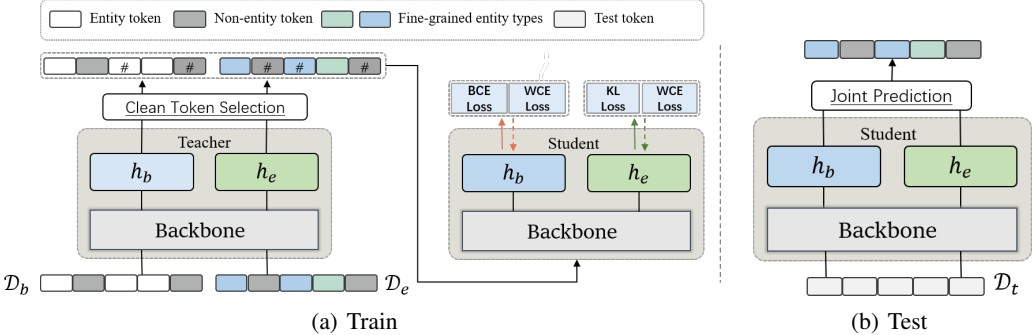

Figure 2: **(a)**: Illustration of decoupled learning paradigm and debiased self-training. **(b)**: Two classification heads are trained independently but make joint predictions when testing. # denotes unselected noisy or invalid tokens.

## 4.1 Decoupled Learning for Distant Supervision

To begin with, we start by introducing the top-tier decomposition which forms the first- and high-level of the taxonomy.

**The Bias Problem.** Let us recall the observation from Figure 1. Extending from it, we draw the following conclusion:

- in Figure 1 (a), we plot the confusion matrix of the true labels against distant labels without distinguishing between fine-grained types. It effectively indicates that the much dominant number of non-entity tokens against the true entity tokens (about ratio **170:23**);

- in Figure 1 (b), we see the noisy distribution of the fine-grained entity-type labels is very much biased and imbalanced. In spite of that the prior work generally omits this problem, we believe this is so vital that the neglect of it would cause an overly high probability of non-entity tokens being picked during the selection process of DSNER. Therefore, only a few entity tokens are selected as clean data for normal NER training;

- Figure 1 (c) further offers supporting evidence. It shows that the previous DSNER frameworks, built with a single-head pathway hardly draft the clean and minority entity tokens. In particular, it samples 436/4,593 (9.49%) MISC tokens and (almost) none I-LOC tokens which are of more value in solving NER.

To sum up, we believe this loss of a few entity tokens being selected — caused by the data bias — is genuinely a huge waste and may have significantly hindered the performance gain. While the prior works generally overlook this problem, in this article, we intend to study it thoroughly and give a solution to resolve it.

**Decomposition of the NER Task.** To overcome the aforementioned data bias, we decompose the dataset $\mathcal{D}$ into two sub-parts, $\mathcal{D} = \mathcal{D}_b \cup \mathcal{D}_e$ where the subscripts of $b$ and $e$ indicate binary- and entity-classification respectively. In effect, we proactively decompose the NER task of single-level prediction into a hierarchical dual that the binary prediction targets entity versus non-entity while the counterpart task is to predict the specific entity category. Notably, the skewed data distribution on $\mathcal{D}$ is also decomposed into two parts on $\mathcal{D}_b$ and $\mathcal{D}_e$, which can be alleviated by separate processing and will not interfere with each other. More specifically, to construct this level of taxonomy, the label set $\tilde{Y}^b \in \mathcal{D}_b$ is binarized. In the meantime, $\tilde{Y}^e \in \mathcal{D}_e$ removes the non-entity tag (as invalid) and only reserves entity type labels.

**Double-head Pathway.** In what follows, to facilitate the training process upon the decomposition, we correspondingly augment a separate pathway to the original single prediction pathway, written as $h = \langle h_b, h_e \rangle$, where the subscripts follow the same naming rule as the previous paragraph. The end of the two pathways are both fully connected layers with different entry numbers: 1 for the $h_b$ and $K$ for $h_e$. The two pathways shared the same encoder, $\phi$, as we introduce in Section 3. A more

detailed illustration is provided in Figure 2 (a). We leave the further textual description of the model architecture and task-data taxonomy details in Appendix A.

**Joint Prediction.** One last piece of utilizing the dual pathway in the framework is the inference procedure. Since either pathway is unable to generate a complete NER prediction independently, we combine them in a jointly predictive fashion. As shown in Figure 2 (b), we form a prediction hierarchy as follows. Given any token $x_i \in \boldsymbol{x}$, the binary pathway $h_b$ offers a probability of that token being a part of an entity denoted as $p_i^b$. On the other hand, the entity pathway generates a vector of specific entity probability $\boldsymbol{p}_i^e$ having $K$ entries. Then the double-head pathway generates a joint probability $\boldsymbol{p}_i = [1 - p_i^b, p_i^b * \boldsymbol{p}_i^e]$, that $1 - p_i^b$ is the probability of being an non-entity. By then, the model can predict each token's specific type, i.e., whether a non-entity or fine-grained entity type.

## 4.2 Clean Token Selection

Following the previous decomposition, the next step we propose is to follow a general noisy-label learning setup — to select the clean set from the noisy dataset. In particular, upon either task/data component in the first level taxonomy, it can be further decomposed down to be a union of the clean and noisy set. Different learning schemes are then applied to these two sets of data and enacted concurrently. In this section, we discuss how the sample selection is devised.

We take the binarized dataset/task to describe the selection mechanism. In hindsight, we assume that the tokens ought to be put into the clean pool when the model's prediction is relatively certain and it matches the provided counterpart label by distant supervision. Formally, given any token $x_i$ drawn from any sentence, denote its distant label as $\tilde{y}_i^b$. During training, we take the model obtained at the current step and feed the input through it attaining a current prediction of $x_i$, dubbed as $\hat{y}_i^b$. Hereby, we can write down the primary selection criterion:

$$\mathcal{D}_b^l = \{(x_i, \tilde{y}_i^b) | \mathbb{I}(\hat{y}_i^b = \tilde{y}_i^b) \wedge (\max(p_i^b, 1 - p_i^b) > \tau)\}$$

Noted, we use $\mathcal{D}_b^l$ to denote the clean set, as superscript $l$ implies "labeled" since the clean tokens are primarily utilized as if they were gold. We simply adopt $\max(p_i^b, 1 - p_i^b)$ as a proximal form of predictive uncertainty on the given sample where $\tau$ is a hyperparameter for the threshold. In most of our experiments, we fix it to $0.95$ without further tuning. Further, the left-out samples — where the token samples do not meet this criterion — are gathered and then form $\mathcal{D}_b^u$ such that $\mathcal{D}_b = \mathcal{D}_b^l \cup \mathcal{D}_b^u$. Here the superscript $u$ means "unlabeled" as we remove the tagged distant supervision labels on them and essentially treat these samples as unlabeled ones. For the finer-grained dataset $\mathcal{D}_e$, we adopt the same strategy, as $\mathcal{D}_e = \mathcal{D}_e^l \cup \mathcal{D}_e^u$.

Notably, the decomposition of the task/data taxonomy is grounded at the token level; namely, the same sentence could contain clean tokens and noisy tokens belonging to different sets for training respectively. Further, the assignment of the taxonomy is dynamic and altered on the fly. This means that at every round of training, DesERT performs token selection based on an inference path conducted by the model trained at the current cycle.

## 4.3 Debiased Self-Training

In the previous step, we resplit the datasets and process on the clean token sets $\mathcal{D}_b^l$ and $\mathcal{D}_e^l$, leaving unselected noisy token sets $\mathcal{D}_b^u$ and $\mathcal{D}_e^u$ remained to be tackled. Since labels for $\mathcal{D}_b^u$ and $\mathcal{D}_e^u$ are likely to be noisy, we remove them and tag these tokens with pseudo-labels via a self-training framework. Notably, the pseudo-labels can be obtained from various sources and we choose a teacher-student architecture that is the same as previous work [10]. Despite that self-training has been proven effective repeatedly [10, 9, 11], we argue that on this occasion it might still be unreliable due to another source of inherent bias.

In practice, the training on the clean and noisy set alternates. In hindsight, the supervised learning on the selected clean set naturally clusters the clean tokens tightly to their respective class centers. By contrast, the noise tokens receive dynamically changing labels, and it is likely that the labels are wrong. This, in theory, may cause the corresponding representation to deviate from the class centers and even oscillate by big magnitude due to its instability assignment during training. Given the imbalance prediction nature of the DSNER task, we empirically find that this kind of bias is

trended to aggravate if no other mechanism is imposed. Further, if we let this bias be untangled, the clean token selection we introduced previously might be severely biased and polluted.

Therefore, to overcome this issue, we employ a worst-case cross-entropy (WCE) loss objective [32]. In a nutshell, the WCE loss adversarially optimizes the unlabeled token representation by matching the worst classifier on labeled data. We call a classifier the worst if it can correctly classify all the labeled data, but is very close to the edge of clusters, resulting in a minimal margin. If we adversarially train the unlabeled tokens to be correctly classified by this worst classifier, their features can be tight to at least one class center. Therefore, the decision hyperplane can be pushed to a lower entropy region, which is known to be beneficial for semi-supervised learning [33]. By regularizing noisy tokens to approach the class centers rather than the decision boundary, the classifier can also generate more stable and higher-quality pseudo-labels. For better understanding, we elaborate our *theoretical insights* of this debiased learning procedure in Appendix B.

Nevertheless, in practice, it is impossible to obtain the truly worst classifier. Hence, proximally, we estimate a possible worst classifier $h_w$ such that it distinguishes the selected tokens correctly while making mistakes on (i)-the unchosen tokens and (ii)-the produced pseudo-labels at previous cycles, as much as possible. Formally, we denote $f^h = h \circ \phi$ and calculate the worst head by,

$$h_w = \arg\max_{h'} \mathcal{L}_U(\hat{\boldsymbol{y}}^u, f^{h'}(\boldsymbol{x}^u)) - \mathcal{L}_L(\tilde{\boldsymbol{y}}^l, f^{h'}(\boldsymbol{x}^l))$$

where the encoder $\phi$ is frozen at this step. $\hat{\boldsymbol{y}}^u = f^h(\boldsymbol{x}^u)$ is the pseudo-label that serves as a proxy to estimate the classification error.

Then, we adversarially optimize the encoder $\phi$ to be correctly classified by the worst-case classifier. The formulation $\mathcal{L}_{wce}$ is given by:

$$\mathcal{L}_{wce}(\phi) = \mathcal{L}_U(\hat{\boldsymbol{y}}^u, f^{h_w}(\boldsymbol{x}^u)) - \mathcal{L}_L(\tilde{\boldsymbol{y}}^l, f^{h_w}(\boldsymbol{x}^l))$$

Noted that we fix the worst-case head $h_w$ and only update the encoder representation. Moreover, since the binary head regards all fine-grained types as an integrated class, optimizing WCE on it may cause a collapse of the representation and then, hinder the model of distinguishing different fine-grained entity types. Therefore, we calculate WCE loss $\mathcal{L}_{e\_wce}$ on the entity-type recognition pathway. Empirically, the noisy token representation gradually approaches to corresponding class center. Consequently, the decision hyperplane is more likely to lie in a lower entropy region, thereby mitigating the impact of self-training bias.

## 4.4 Practical Implementation

**Dual Co-guessing Mechanism.** It has been widely verified that a larger labeled set can greatly boost semi-supervised training. However, in our selection procedure, the size of the clean set is largely restricted by the number of originally clean samples owing to our label-matching constraint. Fortunately, many tokens, while originally noisy, can be accurately predicted by the model as SSL training proceeds. To this end, we develop a co-guessing mechanism that collects well-predicted tokens as a pseudo proxy to enlarge the clean sample set.

Specifically, we train two peer networks having the same architecture, which we denote as $f(\boldsymbol{x}; \theta_1)$ and $f(\boldsymbol{x}; \theta_2)$. Given those easy-to-learn noisy tokens, we postulate that different networks can generate consistent and high-confidence predictions. This gives rise to the following criterion,

$$\{(x_i, \hat{y}_i^{(1)}) | \mathbb{I}(\hat{y}_i^{(1)} = \hat{y}_i^{(2)}) \wedge (\max(\boldsymbol{p}_i^{(1)}) > \tau) \wedge (\max(\boldsymbol{p}_i^{(2)}) > \tau)\}$$

$\hat{y}_i^{(1)}$ and $\hat{y}_i^{(2)}$ are predicted labels from two models, $\boldsymbol{p}_i^{(1)} = \text{softmax}(f_i(\boldsymbol{x}; \theta_1))$ and $\boldsymbol{p}_i^{(2)}$ is same. When it comes to specific sub-datasets $\mathcal{D}_b$ and $\mathcal{D}_e$, the predicted labels should be converted to respective counterparts, and probability is generated by the corresponding pathway.

**Training loss.** The overall training loss is defined as follows:

$$\mathcal{L} = \mathcal{L}_{b\_cls} + \mathcal{L}_{e\_cls} + w * \mathcal{L}_{e\_wce}$$

Here $w > 0$ is a weighting factor. $\mathcal{L}_{b\_cls}$ and $\mathcal{L}_{e\_cls}$ are the classification losses on the two predictive heads respectively. $\mathcal{L}_{e\_wce}$ is the WCE loss on the entity pathway.

Table 1: Main results on five benchmark datasets measured by Precision (P), Recall (R), and F1 scores. We highlight the best overall performance for distant supervision in bold.

| Methods | CoNLL03 | | | OntoNotes5.0 | | | Webpage | | | Wikigold | | | Twitter | | |
|---|---|---|---|---|---|---|---|---|---|---|---|---|---|---|---|
| | P | R | F1 | P | R | F1 | P | R | F1 | P | R | F1 | P | R | F1 |
| **Fully-supervised methods** | | | | | | | | | | | | | | | |
| BiLSTM-CC | 91.35 | 91.06 | 91.21 | 85.99 | 86.36 | 86.17 | 50.07 | 54.76 | 52.34 | 55.40 | 54.30 | 54.90 | 60.01 | 46.16 | 52.18 |
| RoBERTa-base | 89.14 | 91.10 | 90.11 | 84.59 | 87.88 | 86.20 | 66.29 | 79.73 | 72.39 | 85.33 | 87.56 | 86.43 | 51.76 | 52.63 | 52.19 |
| **Distantly-supervised methods** | | | | | | | | | | | | | | | |
| AutoNER | 75.21 | 60.40 | 67.00 | 64.63 | 69.95 | 67.18 | 48.82 | 54.23 | 51.39 | 43.54 | 52.35 | 47.54 | 43.26 | 18.69 | 26.10 |
| LRNT | 79.91 | 61.87 | 69.74 | 67.36 | 68.02 | 67.69 | 46.70 | 48.83 | 47.74 | 45.60 | 46.84 | 46.21 | 46.94 | 15.98 | 23.84 |
| Co-teaching+ | 86.04 | 68.74 | 76.42 | 66.63 | 69.32 | 67.95 | 61.65 | 55.41 | 58.36 | 55.23 | 49.26 | 52.08 | 51.67 | 42.66 | 46.73 |
| JoCoR | 83.65 | 69.69 | 76.04 | 66.74 | 68.74 | 67.73 | 62.14 | 58.78 | 60.42 | 51.48 | 51.23 | 51.35 | 49.40 | 45.59 | 47.42 |
| NegSampling | 80.17 | 77.72 | 78.93 | 64.59 | **72.39** | 68.26 | 70.16 | 58.78 | 63.97 | 49.49 | 55.35 | 52.26 | 50.25 | 44.95 | 47.45 |
| BOND | 82.05 | 80.92 | 81.48 | 67.14 | 69.61 | 68.35 | 67.37 | 64.19 | 65.74 | 53.44 | 68.58 | 60.07 | 53.16 | 43.76 | 48.01 |
| SCDL | **87.96** | 79.82 | 83.69 | **67.49** | 69.77 | 68.61 | 68.71 | 68.24 | 68.47 | 62.25 | 66.12 | 64.13 | **59.87** | 44.57 | 51.09 |
| **Ours*** | 86.23 | 87.28 | 86.75 | 66.38 | 72.08 | 69.11 | 71.52 | **72.97** | 72.24 | 60.77 | 68.10 | 64.23 | 56.44 | **48.38** | 52.10 |
| **Ours*(finetune)** | 86.41 | **87.49** | **86.95** | 66.63 | 71.92 | **69.17** | **72.48** | **72.97** | **72.73** | **62.87** | **69.42** | **65.99** | 57.65 | 47.80 | **52.26** |

**Post-hoc Entity Pathway Finetuning.** In previous steps, we train the double-head pathway separately as both pathways can be unreliable due to the label noise. Over the course of training, we observe the binary pathway learns well thanks to abundant training samples. On the contrary, the entity pathway still faces a rather challenging condition caused by inefficient samples, label noise, and more categories. To relieve this problem, we propose to post-hoc finetune the entity pathway by constructing a more refined dataset via the binary pathway. In specific, after the training procedure, we freeze the binary pathway $h_b$ and collect a new clean entity dataset by,

$$\mathcal{D}_e^l = \{(x_i, \hat{y}_i^e) | \mathbb{I}(\hat{y}_i^b = 1) \wedge \mathbb{I}(\hat{y}_i^e = \tilde{y}_i^e)\}$$

That is, while still anticipating selecting clean samples, we remove those tokens predicted as non-entity by $h_b$, which produces a more refined clean sample set than before. Lastly, we finetune the entity pathway with this clean dataset for a few epochs to boost the performance.

## 5 Experiments

In this section, we present our main experimental results to show the effectiveness of our method. More empirical setups and results are reported in Appendix C.

### 5.1 Setup

**Dataset.** We evaluate our framework on five widely-used named entity recognition benchmark datasets in the English language: (1) **CoNLL03** [34] is collected CoNLL 2003 Shared Task, consisting of 1,393 English news articles and four entity types; (2) **OntoNotes5.0** [35] contains 1.6 million words annotated by 18 entity types from multiple domains, including broadcast conversation, Web data, and P2.5 data; (3) **Webpage** [36] collects 783 entities belonging to the four types like CoNLL03, which are from 20 webpages including personal, academic, and computer-science conference homepages. (4) **Wikigold** [37] comprise 39,007 Wikipedia article tokens from a 2008 English dump with four CoNLL03 named entity tags; (5) **Twitter** [38] is an open-domain NER dataset coming from the WNUT 2016 NER shared task, which consists of 2,400 tweets (a total 34k tokens) with 10 entity tags. To obtain a distant supervision scheme, we follow previous works [10] to re-annotate all the training corpus by matching entity types in external knowledge bases including Wikidata corpus and gazetteers extracted from multiple online sources.

**Baselines.** We compare the proposed framework with a total of seven current *distantly-supervised* methods: (1) **AutoNER** [7] trains a fuzzy LSTM-CRF network to handle tokens with multiple possible labels, using a Tie or Break scheme; (2) **LRNT** [17] employs Paritial-CRF for sequence labeling and training with high-quality sentences given by high annotation confidence and coverage; (3) **Co-teaching+** [39] is a robust sample selection-based algorithm, which only updates the dual networks by prediction disagreement data; (4) **JoCoR** [40] adopts dual networks to calculate a

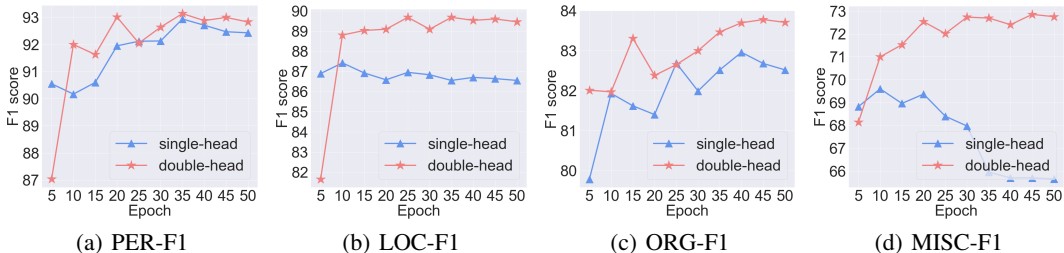

Figure 3: The F1-scores of DesERT with/without double-head pathway on four entity types, which have 11.1k/8.3k/10.0k/4.6k tokens respectively (from left to right).

joint loss with co-regularization; (5) **NegSampling** [41] handles incomplete annotations by negative sampling; (6) **BOND** [10] is a self-training paradigm that selects reliable clean tokens for training by a teacher-student network with early stopping; (7) **SCDL** [9] jointly trains two teacher-student networks to iteratively perform label refinery in a mutually beneficial manner. Besides, we select two *fully-supervised* methods trained with clean annotations to show the performance upper bound: (1) **BiLSTM-CNN-CRF (BiLSTM-CC)** [3] adopts bi-directional LSTM with character-level CNN to generate token embeddings, followed by a CRF layer to predict the most likely label sequence; (2) **RoBERTa-base** [5] adopts a pre-trained RoBERTa-base model as the encoder.

**Implementation details.** We adopt two pre-trained language models RoBERTa-base and DistilRoBERTa as the backbone. Inspired by [10], we further maintain a teacher network that is an exponential moving average of the main model. Thereafter, the data split, pseudo-labeling, loss calculation, and co-guessing operators are done by the prediction of the teacher network; see Appendix C.4 for detailed descriptions. We follow [10, 9] to tune most hyperparameters. Specifically, we train the networks for 50 epochs with a few epochs of warm-up, followed by 2 epochs of finetuning. The training batch size is set as 16 on four datasets, except 32 on OntoNotes5.0. The learning rate is fixed as $\{1e^{-5}, 2e^{-5}, 1e^{-5}, 1e^{-5}, 2e^{-5}\}$ for these datasets respectively. The confidence threshold parameter $\tau$ is tuned by 0.9 for Wikigold while 0.95 for others. The co-guessing is performed from the $k$-th epoch, which we set to $\{6, 40, 35, 30, 30\}$ respectively. For finetuning, the learning rate is one-tenth of the original one. Once the training ends, we choose the first student model for predictions as the final results. We run all experiments for three random trials and record the mean results.

## 5.2 Main Results

We measure the performance of all methods by Precision (P), Recall (R), and F1 scores. As shown in Table 1, our proposed method DesERT achieves state-of-the-art performance compared with other baselines on five benchmark datasets. Specifically, on the CoNLL03 and Webpage datasets, DesERT outperforms the best baseline methods by 3.26% and 4.26% in terms of F1 score. The performance is even close to the best fully-supervised baseline (i.e. RoBERTa-base) on Webpage and Twitter, which suggests better adaptation to distant supervision. In addition, DesERT displays significantly higher recall on almost all datasets, demonstrating a strong ability to identify entities from text. It also maintains moderate precision so as to attain extraordinary F1 scores. Finally, the model after post-hoc finetuning shows obviously better performance. It demonstrates that a well-trained binary head can help the fine-grained head to better distinguish entity types, proving the superiority of the task decomposition formula.

## 5.3 Ablation Study

To explore the effectiveness of each component in DesERT, we conduct an ablation study on the CoNLL03 dataset without the finetuning stage. As shown in Table 9, we test the following cases: (1) remove the WCE loss; (2) do not perform dual networks co-guessing; (3) replace the double-head pathway with a single-head pathway.

It can be seen from the ablations that the removal of any component leads to performance degradation, while the model can easily obtain the best F1 score under the combined effects of three components.

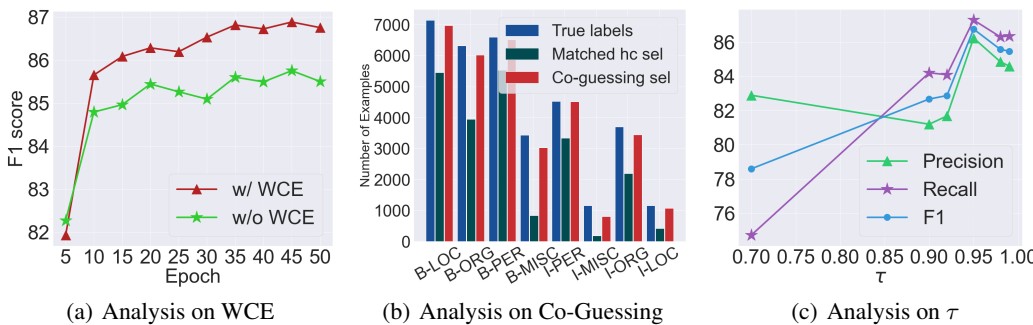

|       (a) Analysis on WCE       |     (b) Analysis on Co-Guessing     |       (c) Analysis on $\tau$       |

Figure 4: (a) The F1 curves of DesERT with/without WCE loss. (b) The distribution of the true labels and selected labels with/without co-guessing. (c) The parameter study of different confidence thresholds $\tau$ on the CoNLL03 dataset.

Specifically, discarding the co-guessing mechanism will cause the maximum performance descent of 3.31% on F1 scores, which indicates the benefits of using co-guessing to correct distant labels for training. As the core mechanism of DesERT, the double-head pathway can also improve the F1 score by 2.21% compared with only a single-head pathway, indicating that the idea of task-data decomposition is more suitable for distant supervision. Finally, WCE loss provides an effective performance gain as a debiased add-on. We additionally note that Precision and Recall metrics are also optimal, which proves the model can accurately separate entity and non-entity tokens as well as identify relatively complete entities.

## 5.4 Further Analysis

**Efficacy of decoupled learning.** To validate the efficacy of decoupled learning, we further test the F1 score of each entity type in two cases on the CoNLL03. First, by comparing the F1 scores of four entity classes shown in Figure 3, it can be seen that the conventional single-head model demonstrates inferior results on all entity types. More importantly, we found that the two models show relatively close behavior in identifying entity classes with sample-rich tokens like PER. In contrast, the performance gap in minority categories is large.

Table 2: Ablation study on CoNLL03 dataset.

| Ablations | CoNLL03 | | |
|---|---|---|---|
|  | **P** | **R** | **F1** |
| **Ours*** | **86.23** | **87.28** | **86.75** |
| w/o WCE | 85.04 | 85.98 | 85.50 |
| w/o co-guessing | 82.34 | 84.57 | 83.44 |
| w/o double-head | 84.18 | 84.90 | 84.54 |

For example, the F1 score on the MISC entities of the double-head pathway is 7.11% higher than that of the single-head pathway (Figure 3 (d)), and the gap for LOC entities is 2.91% (Figure 3 (b)). Therefore, the design of a double-head pathway or the idea of task decomposition can well adapt to DSNER and demonstrate stronger entity recognition capability for minority entity classes.

**Efficacy of debiased self-training.** We study the effectiveness of WCE loss for the DesERT framework. We test the cases with and without adopting WCE loss on the CoNLL03 dataset, and the results are shown in Figure 4 (a). It can be found that WCE loss is more likely to serve as an effective add-on in DesERT. It should be noted that WCE displays high-performance growth in early epochs, which alleviates the model bias towards unpurified labels in the early stage.

**Efficacy of co-guessing.** Besides, we explore the ability of the co-guessing mechanism to select extra tokens. In Figure 4 (b), we compare the selected token distribution with and without the co-guessing operation. Without co-guessing, a huge gap occurs between the selected tokens per class and the real distribution, e.g., more than 3,000 tokens are not selected on the already scarce class MISC. The reason is that we enforce the clean tokens to have a matched prediction on given labels. This severely impairs the recognition performance of minority entities. With our co-guessing mechanism, we observe the clean sample set is significantly enlarged and the distribution of selected entity label distribution also approaches that of true labels. Combined with the fact in Table 9 that co-guessing does indeed improve performance, the effectiveness of co-guessing is clearly validated.

Table 3: Performance of DesERT with different sources of distant labels on CoNLL03.

| Supervision | Unsupervised | | ChatGPT Labels | | KB Labels | | Hybrid Labels | |
|---|---|---|---|---|---|---|---|---|
| Model | ChatGPT | ChatGPT-A | SCDL | DesERT | SCDL | DesERT | SCDL* | DesERT* |
| Precision | 68.95 | 79.11 | 68.39 | 81.91 | **87.96** | 86.23 | 83.87 | 87.24 |
| Recall | 64.16 | 63.13 | 72.74 | 77.38 | 79.82 | 87.28 | 85.50 | **88.93** |
| F1 | 66.47 | 70.22 | 70.50 | 79.58 | 83.69 | 86.75 | 84.67 | **88.08** |

**Efficacy of threshold parameter $\tau$.** We study the effect of the important hyperparameter $\tau$ used in clean sample selection and dual networks co-guessing on the CoNLL03 dataset. Specifically, we vary the value of $\tau$ in the range 0.7 to 0.99, and other parameters are kept as default. Then we validate DesERT's performance by applying different $\tau$ values, as shown in Figure 4 (c). We can see that the model's performance is relatively insensitive in the $\tau$ varies from 0.95 to 0.99, but drops a lot when the $\tau$ value is lower than 0.95. And DesERT achieves the best results with $\tau$ fixed at 0.95 on the CoNLL03. In practice, we suggest setting a high confidence threshold $\tau$.

## 5.5 Distant Supervision from Large Language Models

Recently, large language models (LLMs), including GPT-3 [42], ChatGPT, and GPT-4[3], have largely revolutionized the NLP landscape. Thanks to their emerging abilities like in-context learning (ICL) [43] and chain-of-thought [44], LLMs demonstrate remarkable zero-shot learning performance in a wide range of downstream NLP tasks. Despite the promise, some recent studies [45] have shown that LLMs are still legs behind the fine-tuned small language models in many NLP applications including NER. To deal with this problem, we extend the DSNER formulation and design a novel in-context learning algorithm that exploits self-generated text-tag pairs to generate distant labels. Moreover, we modify our original algorithms to fully use hybrid labels including ChatGPT-generated labels and original knowledge-base-generated labels (KB labels). We refer the readers to Appendix C.1 for a detailed description of our prompt design, data generation process, and methodology.

Empirically, we provide a series of experiments to verify the feasibility of utilizing ChatGPT for distant supervision. In particular, we compare DesERT with three algorithms: (1) **ChatGPT**: we employ the ChatGPT model to produce zero-shot tagging on testing data; (2) **ChatGPT-A**: we employ ChatGPT to generate a set of text-tag pairs and use it for few-shot ICL on testing data; (3) **SCDL**: the most competitive baseline in our main Table. Table 4 lists our full results, we have the following conclusions: (1) ChatGPT does indeed demonstrate inferior results even compared with distantly-supervised SLMs; (2) simply training current best-performing DSNER methods on ChatGPT supervision can be inferior since they are particularly designed for KB labels; (3) ChatGPT can be a complementary source of distant supervision to existing KB labels, and our modified DesERT (dubbed DesERT*) still obtains best performance when handling hybrid labels. Overall, we firmly believe that DSNER remains a pivotal research pursuit in the LLMs era. Looking forward, we envision LLMs serving as a novel source of distant supervision, facilitating the distillation of distantly-supervised SLMs, which has enhanced accuracy and computational efficiency.

## 6 Conclusion

In this paper, we propose a novel framework DesERT for NER under distant supervision. We first find that existing self-training methods suffer from two types of biases, i.e., a skewed data distribution in distant supervision and an inherent bias to assign erroneous pseudo-labels. To mitigate both biases, we propose a novel task/data taxonomy to decompose the original task, alongside a debiased self-training to improve the quality of pseudo-labels. Extensive experiments on other standard benchmarks clearly validate the effectiveness of DesERT. Furthermore, we supplement a new ChatGPT-based DSNER benchmark and show that DesERT still obtains the best performance. We hope our work can draw more attention from the community to the bias issues in the distantly-supervised NER tasks.

---

[3]https://openai.com/blog/chatgpt

## Acknowledgments and Disclosure of Funding

This work is majorly supported by the National Key Research and Development Program of China (No. 2022YFB3304100), and in part by the NSFC under Grants (No. 62206247). Junbo Zhao also thanks the sponsorship by CAAI-Huawei Open Fund and the Fundamental Research Funds for the Central Universities (No. 226-2022-00028).

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

# A    Dataset-Task Taxonomy

## A.1    Dataset Taxonomy

We provide Figure 5 to show the dataset taxonomy in the overall training workflow. We first split the initial dataset $\mathcal{D}$ into two sub-datasets $\mathcal{D}_b$ and $\mathcal{D}_e$ through decoupled learning to mitigate the distributional bias (first level split). Next, to process respective noise in $\mathcal{D}_b$ and $\mathcal{D}_e$, we conduct a selection and self-training framework. We divide the two parts into the clean set and noisy set by clean token selection (second level split), then perform a standard classification on each clean set. As for noisy token sets, we adopt debiased self-training to yield pseudo-labels for further training.

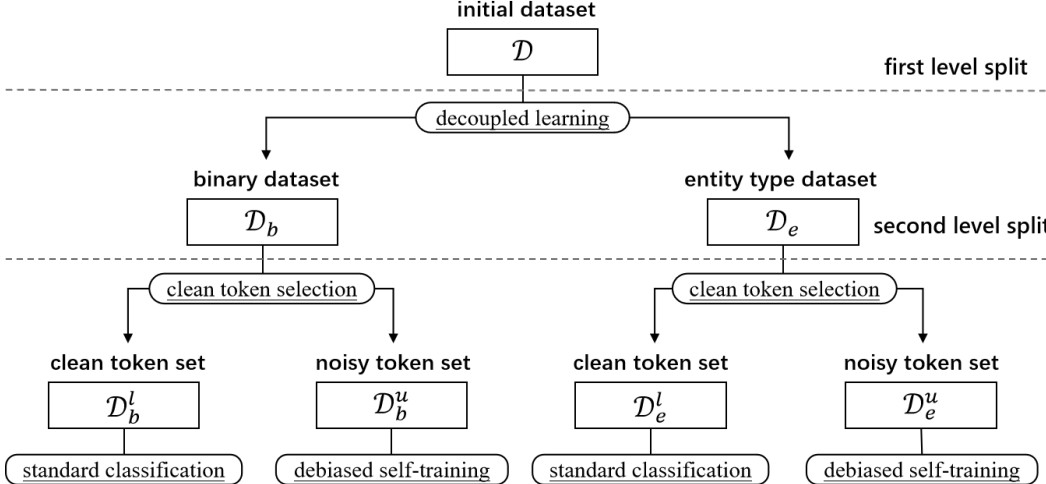

Figure 5: Illustration of dataset taxonomy.

## A.2    Task Taxonomy

DesERT modifies the basic NER model architecture with the double-head pathway, yet reserves a shared pre-trained language model encoder such as RoBERTa-base denoted by $\phi$. Given any sentence $\boldsymbol{x}$ with its binarilized labels $\boldsymbol{y}^b$ and entity type labels $\boldsymbol{y}^e$, $(\boldsymbol{x}, \boldsymbol{y}^b) \in \mathcal{D}^b$ and $(\boldsymbol{x}, \boldsymbol{y}^e) \in \mathcal{D}^e$. $\boldsymbol{x}$ is first fed to the PLM encoder $\phi$ and we take the last hidden layer output of $\phi$ as finally embeddings $\phi(\boldsymbol{x})$. Then the double-head pathway $h_b$ and $h_e$ take $\phi(\boldsymbol{x})$ as input to yield respective predictions. The binary pathway $h_b$ generates the probablity of being entity tokens, $\boldsymbol{p}^b = \text{sigmoid}(h_b \circ \phi(\boldsymbol{x})) = [p_1^b, ..., p_n^b]$, then take $\boldsymbol{p}^b > 0.5$ as predicted binary labels $\hat{\boldsymbol{y}}^b$. While the entity pathway $h_e$ offers fine-grained entity type probablities $\boldsymbol{p}^e = \text{softmax}(h_e \circ \phi(\boldsymbol{x})) = [\boldsymbol{p}_1^e, ..., \boldsymbol{p}_n^e]$, where each $\boldsymbol{p}_i^e$ has $K$ entries. We take $\arg\max(\boldsymbol{p}^e)$ as predicted entity type labels $\hat{\boldsymbol{y}}^e$, and note that non-entity tokens are tagged with invalid labels. Finally, any standard classification loss can be calculated on the double-head pathway. We also refer the reader to Figure 2 for visualized illustration.

# B    Theoretical Insights of Debiased Semi-supervised Learning

Though there is no available theory on why the worst cross-entropy (WCE) [32] works, we would like to provide the following (relatively) theoretical insights that may help the readers to perceive our approach better.

Notably, the self-training bias is mainly caused by noisy tokens approaching the decision boundaries, whose pseudo-labels keep changing. To this end, we optimize WCE to learn compact token clusters for reducing wrong pseudo-label assignments. To see this, we start from the following simplified example to show that WCE does indeed concentrate the token representation.

**Assumption.**    Consider a binary classification problem, as the simplest form of DSNER. Denote the input variable by $X$ and the output variable by $Y$ from binary labels $\{+1, -1\}$. The labeled data $\mathcal{D}^l$

are sampled from $X|Y = +1 \sim U(\mathcal{B}(u, r))$ and $X|Y = -1 \sim U(\mathcal{B}(v, r))$. Here $\mathcal{B}(u, r)$ denotes a spherical ball with center $u$ and radius $r$. $U$ denotes uniform distributions. The unlabeled sample $\mathcal{D}^u$ are all from $Y = +1$ but uniformly distributed inside the $\mathcal{B}((u + v)/2, r')$. Three balls have no intersection. Finally, we assume the maximum margin classifier is used, which is a hard proxy of the cross-entropy loss.

**Derivation Sketch.** At first glance, it is obvious that the optimal classifier on $D^l$ is $f = (u + v)/2$ which misclassifies half of the examples in $D^u$. The worst classifier, however, amounts to be $f^w = (u + v)/2 - r'$ which perfectly classifies $D^l$ but possesses the most side-way decision boundary. Next, we optimize the feature extractor $\phi$ to match the worst decision boundary by $\min_\phi L_U(y, f^w)$. This is to say, with ideally known labels, the unlabeled ball $\mathcal{B}((u + v)/2, r')$ converges to $\mathcal{B}(f = (u + v)/2 - r', r')$. With full samples $X|Y = +1$ getting closer, the classifier achieves better generalization with *compact clusters* and low-entropy decision boundaries.

In practice, since the true labels are unknown, we use pseudo-labels as a proxy since most unlabeled data are assigned true labels. So, the representation will be partially concentrated.

**Empirical Covariance.** We conduct experiments on CoNLL03 to show the covariance of data to their class centers. When DesERT is run without WCE, the average covariance amongst classes is 0.0085. With WCE, the average covariance amongst classes becomes 0.0056. Thereafter, we can conclude that WCE indeed concentrates the tokens and mitigates the self-training bias.

## C  Additional Experimental Setups and Results

In what follows, we show more experimental details and results. In section C.1, we report more empirical results, including an interesting series of experiments where *additional distant supervision* comes from large language models like ChatGPT. In section C.3, we provide more details on our experiments and implementation.

### C.1  Results with Additional Distant Supervision from ChatGPT

Recently, large language models (LLMs), including GPT-3 [42], ChatGPT, and GPT-4[4], have largely revolutionized the NLP landscape. Thanks to their emerging abilities like in-context learning (ICL) [43] and chain-of-thought [44], LLMs demonstrate remarkable zero-shot learning performance in a wide range of downstream NLP tasks. Despite the promise, some recent studies [45] have shown that LLMs are still legs behind the fine-tuned small language models in many NLP applications.

Motivated by this, we conduct experiments to show the zero-shot performance of ChatGPT on the NER problem. In Table 4, we observe that ChatGPT does indeed demonstrate inferior results even compared with distantly-supervised SLMs. Therefore, a question arises: *how can LLMs better support NER with minimal human annotation?* Among the numerous potential solutions, we propose a natural extension of the conventional distantly-supervised NER problem. This extension considers LLM's predictions on the training set as additional distant supervision. To achieve this, we present a new tagging scheme for NER and modify our algorithm to accommodate multiple sources of distant supervision. Subsequently, we provide a detailed elaboration on the aforementioned aspects.

#### C.1.1  Tagging by Auto-Demonstration

Instead of directly performing zero-shot testing, we introduce a new DSNER paradigm that generates distant labels by LLMs for *unsupervised training data* to improve the SLMs' performance. To generate distant labels, a straightforward solution is to send the raw training texts to the LLMs and ask them to output all the entities along with their types. However, we find such a naive strategy fails to achieve satisfactory NER performance. To mitigate this problem, we design a novel in-context learning algorithm that exploits self-generated text-tag pairs to guide the tagging process.

**Automatic Text-Tag Pair Generation.** Our ultimate goal is to perform few-shot in-context learning that better guides the LLMs to locate the entities and output their types. To achieve this,

---

[4]https://openai.com/blog/chatgpt

Figure 6: An example prompt for automatic demonstration data generation.

we may assume a set of demonstration text-tag pair samples are available. Besides, this set ought to be diverse and representative enough to guide in-context learning. However, the training samples are unsupervised and cannot be utilized directly. To address this problem, we propose to generate sentences by ChatGPT itself, while ensuring the diversity of the generated results to cover all entity types. Additionally, we randomly retrieve unlabeled samples from the training set and employ ChatGPT to automatically generate a comprehensive set of sentences, including their corresponding entity tags. An example prompt is shown in Figure 6.

**Few-Shot In-Context Learning for NER Tagging.** After that, we ask the ChatGPT to tag the whole training set by using its self-generated demonstrations. However, directly feeding all the generated samples for ICL may exceed contextual limits and incur high computational costs. To remedy this problem, before feeding the true query sentence, we retrieve the top-$k$ generated samples by the cosine similarity. Empirically, we exploit the BERT embedding for similarity calculation, which performs generally better than ChatGPT embedding. Finally, we instruct ChatGPT to output the NER tags for the entire unsupervised training set. An example prompt is shown in Figure 7.

### C.1.2 Modification of DesERT for Multi-Source Distant Labels

One may directly use the ChatGPT labels to train the DSNER models. However, the original knowledge base-driven distant supervision is also a free lunch for DSNER. and can be further incorporated. Notably, such a hybrid distant label from multiple sources problem has never been touched in the NER community. Fortunately, our DesERT algorithm can well address such hybrid labels with only a few modifications. Assume we are given knowledge base-driven labels $y_{kb}$ (KB Labels) and ChatGPT Labels $y_{cg}$ for a token $x$, where we slightly abuse the subscript to distinguish these two labels. To warm up the model, we calculate a mean soft label by,

$$\hat{y}^{mean} = (\text{OneHot}(y^{kb}) + \text{OneHot}(y^{cg}))/2$$

Therefore, the model fits equal confidence on these two labels when a token receives two disagreed labels. After that, we develop a modified selection protocol. Take the binary head as an example, we receive a set of transformed distant labels $\tilde{Y}^b = \{\tilde{y}^b_{kb}, \tilde{y}^b_{cg}\}$ for each token and then perform the token selection by,

$$\mathcal{D}^l_b = \{(x, \tilde{Y}^b) | \mathbb{I}(\hat{y}^b \in \tilde{Y}^b) \wedge (\max(p^b, 1 - p^b) > \tau)\}$$

In other words, we regard either distant label as a candidate of the ground truth. Once there is one label in this label set that exhibits high confidence, we regard it as a clean token. Finally, we run the

Figure 7: An example prompt for few-shot in-context tagging for a *query sentence*. The demonstrations are automatically generated and selected.

DesERT algorithm without any further modification. Notably, these can be easily extended to more sources of distant supervision, e.g., when there is more than one LLM available.

### C.1.3   Experimental Results with ChatGPT Supervision

Our experiments are conducted on the CoNLL03 dataset. Specifically, we generate a total of 100 automatically generated samples with the help of 10 demonstration training samples. When tagging training data, we select the 10 most similar generated samples for ICL. In Figure 8, we plot the confusion matrix of the ChatGPT labels. It can be observed that ChatGPT supervision demonstrates similar trends to KB labels and is still biased. Nevertheless, ChatGPT labels have three main characteristics: (1) they classify far more non-entity tokens as an entity; (2) they produce more

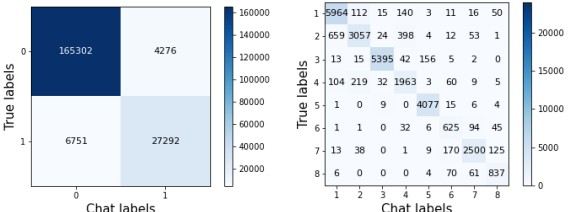

Figure 8: **Left**: Confusion matrix of true labels and ChatGPT labels on CoNLL03. **Right**: The confusion matrix displays noise among true entity-type labels in ChatGPT labels.

balanced clean tokens; (3) the confusion patterns on fine-grained entity types are different than KB labels.

In Table 4, we report the results of DesERT when it faces different sources of distant supervision signals. In particular, we compare DesERT with three baselines: (1) **ChatGPT**: we employ the ChatGPT model to produce zero-shot tagging on testing data; (2) **ChatGPT-A**: we employ ChatGPT to generate a set of text-tag pairs and use it for few-shot ICL on testing data; (3) **SCDL**: the most competitive baseline in our main Table. From Table 4, we have the following observation:

- ChatGPT-A is much better than vanilla ChatGPT, verifying the superiority of our automatic demonstration process. But, ChatGPT and ChatGPT-A underperform other DSNER algorithms on the testing set. Though ChatGPT is a wonderful general-purpose LLM, we may draw the same conclusion as [45] that fine-tuned SLMs still play an important role in NLP.

- Given ChatGPT labels, both SCDL and DesERT underperform their counterparts when supervised by KB labels. We postulate that current DSNER algorithms are particularly designed for KB-based supervision and thus can not fully handle such new sources of labels.

Table 4: Performance of DesERT with different sources of distant labels on CoNLL03.

| Supervision | Unsupervised | | ChatGPT Labels | | KB Labels | | Hybrid Labels | |
|---|---|---|---|---|---|---|---|---|
| Model | ChatGPT | ChatGPT-A | SCDL | DesERT | SCDL | DesERT | SCDL* | DesERT* |
| Precision | 68.95 | 79.11 | 68.39 | 81.91 | **87.96** | 86.23 | 83.87 | 87.24 |
| Recall | 64.16 | 63.13 | 72.74 | 77.38 | 79.82 | 87.28 | 85.50 | **88.93** |
| F1 | 66.47 | 70.22 | 70.50 | 79.58 | 83.69 | 86.75 | 84.67 | **88.08** |

- Our proposed DesERT algorithm consistently outperforms baselines on different supervised signals. In particular, when trained with hybrid labels, the modified DesERT (DesERT*) improves the KB label-trained counterpart by $+$**1.33** F1 score. It suggests that distant supervision from LLMs does indeed brings helpful information for the DSNER task.

In summary, our work makes the first attempt to employ LLMs to generate distant supervision. Moreover, our DesERT algorithm can be easily extended to learn from multi-source distant labels and demonstrates improved performance.

## C.2 More Experimental Results

Table 5: The sensitivity of $k$ without fine-tuning.

| Datasets | $k =15$ | 20 | 25 | 30 | 35 | 40 |
|---|---|---|---|---|---|---|
| CoNLL03 | 86.72 | 86.52 | 86.78 | 86.82 | 86.48 | 86.18 |
| Twitter | 52.22 | 52.15 | 52.21 | 52.10 | 52.44 | 51.54 |

### C.2.1 Sensitivity of $k$

When tuning the co-guessing epoch $k$, we find that setting $k = 6$ has led to good performance for CoNLL03. For other datasets, we found that it is proper to tune it among $k \in \{30, 35, 40\}$. In Table 5, we additionally present the parameter sensitivity analysis of $k$. It can be observed that DesERT is not very sensitive to the $k$ value.

| Ablations | CoNLL03 | | | OntoNotes5.0 | | | Webpage | | | Wikigold | | | Twitter | | |
|---|---|---|---|---|---|---|---|---|---|---|---|---|---|---|---|
| | P | R | F1 | P | R | F1 | P | R | F1 | P | R | F1 | P | R | F1 |
| w/o Binary WCE | **86.23** | **87.28** | **86.75** | 66.38 | 72.08 | 69.11 | **71.52** | 72.97 | **72.24** | **60.77** | **68.10** | **64.23** | **56.44** | **48.38** | **52.10** |
| w/ Binary WCE | 84.29 | 86.38 | 85.32 | **66.43** | **72.14** | **69.17** | 70.78 | **73.65** | 72.18 | 59.64 | 67.34 | 63.25 | 56.32 | 48.06 | 51.86 |

Table 6: Performance of DesERT with binary WCE on five benchmark datasets measured by Precision (P), Recall (R), and F1 scores.

### C.2.2 The Effect of Debiased Training on the Binary Head

In this section, we further investigate whether the binary head can benefit from the debiased self-training procedure. In particular, we optimize the worst-case cross-entropy loss $\mathcal{L}_{b\_wce}$ on the binary head pathway as well. The final objective is,

$$\mathcal{L} = \mathcal{L}_{b\_cls} + w * \mathcal{L}_{b\_wce} + \mathcal{L}_{e\_cls} + w * \mathcal{L}_{e\_wce}$$

In Table 6, we can observe that training with binary WCE generally lowers the F1 scores on benchmark datasets. We speculate that the reason is the binary classification head does not have adequate categorical information for training the representation. Moreover, if we apply WCE on the binary head, tokens from different fine-grained entities might be condensed into a large cluster. This may hinder the model of distinguishing different fine-grained entity types.

### C.2.3 The Standard Deviation of Main Results

Our implementation and configurations closely adhere to the codes of existing algorithms [9]. Consequently, our baseline results align with those presented in the respective source papers. In

Table 7: The standard deviation of DesERT on benchmark datasets.

| Methods | CoNLL03 | OntoNotes5.0 | Webpage | Wikigold | Twitter |
|---|---|---|---|---|---|
| DesERT | 0.22 | 0.43 | 0.40 | 0.46 | 0.32 |
| DesERT (finetune) | 0.17 | 0.20 | 0.22 | 0.30 | 0.17 |

Table 8: The standard deviation of DesERT on benchmark datasets.

| Methods | SNIPS (20% noise) | SNIPS (30% noise) | ATIS (20% noise) | ATIS (30% noise) |
|---|---|---|---|---|
| DesERT | 94.29 | 93.51 | 95.15 | 94.73 |
| RoBERTa | 85.03 | 77.09 | 90.89 | 90.05 |

order to maintain uniformity in formatting, we have refrained from reporting the variance, due to the absence of such information in the cited references. In Table 7, we present the standard deviation values corresponding to our initial results. It can be shown that our variances are not substantial. In view of the notable performance gap (average $+2.22$ F1), we believe that our algorithm has shown statistically significant improvement.

### C.2.4 Experiments on Other Sequential Labeling Tasks

In our main content, we discussed the superiority of DesERT on the DSNER task. However, our method is actually a generic framework and can be applied to other sequential labeling tasks. To see this, we follow [46] on the sequence labeling benchmarks SNIPS and ATIS in a quest to make the comparison fair. We followed the literature on noisy label learning and injected 20% and 30% synthetic noise. The results in F1 are shown in Table 8. It says that DesERT performs way better than its counterparts.

Further, we conducted a simple experiment on the bioinformatics dataset FLIP that predicts the second structures of the proteins [47]. We injected 20% noise into synthetic datasets. Owing to the time limits, we adapted the decoupled training and debiased SSL modules to the benchmark from ProtTrans. In particular, we used a 30-layered ProtBERT model as the backbone and only fine-tuned the top 5 layers due to

Table 9: Valid accuracy on the FLIP dataset.

| Methods | Accuracy |
|---|---|
| ProtBERT (Supervised) | 80.83 |
| ProtBERT (20% noise) | 69.03 |
| DesERT (20% noise) | 70.38 |

limited computation sources. We found that ProtBERT is quite robust, and thus, we only finetune its top 5 layers. However, DesERT still shows some benefits in improving the robustness. These results prove the superiority of DesERT in handling weakly-supervised sequential tagging tasks.

### C.3 More Implementation Details

In this section, we provide more experimental details for a better understanding of our training process and also to ensure the reproducibility.

### C.3.1 Computation Resources

All experiments are conducted on a workstation with 8 NVIDIA RTX A6000 GPUs. It takes about $\{8, 72, 0.5, 1, 2\}$ hours for training on five benchmarking datasets (ordered as in Table 1) with one single GPU. We adopt the Huggingface Transformer library for the RoBERTa-base (125M parameters) and DistilRoBERTa-base (66M parameters) models: https://huggingface.co/transformers/. We run all the experiments three times and report the mean results.

### C.3.2 Datasets Details

The data statistics of five benchmarking NER datasets are shown in Table 10.

Table 10: The statistics of five datasets, show the number of entity types and the number of sentences in the Train/Dev/Test set.

| Dataset | Types | Train | Dev | Test |
|---------|-------|-------|-----|------|
| CoNLL03 | 4 | 14,041 | 3,250 | 3,453 |
| OntoNotes5.0 | 18 | 115,812 | 15,680 | 12,217 |
| Webpage | 4 | 385 | 99 | 135 |
| Wikigold | 4 | 1,142 | 280 | 274 |
| Twitter | 10 | 2,393 | 999 | 3,844 |

### C.4 More Implementation Details

**Tagging scheme for NER.** As for the tagging scheme, we follow the classic BIO format. To be specific, the first token of an entity mentioned with type X is tagged as B-X while the remaining tokens inside that entity are labeled as I-X, and the non-entity tokens are annotated as O. Such a scheme is more difficult than a simple IO format, especially for distant supervision.

**Clean token selection for the binary pathway.** In general, the precision of entity labels is relatively high, e.g., about $97.96\%(23,649/24,141)$ in the CoNLL03 dataset. That indicates us most distantly-labeled entity tokens are real entities if omit fine-grained entity types. Therefore, when performing clean token selection on the binary pathway, it only selects non-entity tokens by the matched and high-confidence strategy while including all tokens labeled as entities.

**Soft label for the entity pathway.** When training on the selected clean token set, the binary pathway regards distant labels as true labels. However, we discard the distant hard labels but adopt the teacher model's output logits to derive soft labels [48] for the entity pathway, given by:

$$\hat{y}_{i,j}^s = \frac{\boldsymbol{p}_{i,j}^2/\sum_i \boldsymbol{p}_{i,j}}{\sum_{j'}(\boldsymbol{p}_{i,j}^2/\sum_i \boldsymbol{p}_{i,j'})}$$

where $\boldsymbol{p}_{i,j} = \text{softmax}(f_{i,j}(\boldsymbol{x};\theta_t))$, is the probability of $i$-th token belonging to class $j$ in sentence $\boldsymbol{x}$, then calculate a Kullback-Leibler divergence loss. Because soft labels usually preserve sufficient information and encourage a more balanced assignment of target labels.

**The implementation of teacher-student newtork.** When splitting the clean token set and the noisy token set, we let the teacher model select respective clean tokens to train the student model. Specifically, the teacher model's double-head pathway filters reliable clean tokens independently following the previous defined criterion. Then the double-head of the student model is trained with selected clean tokens and corresponding labels. The teacher model parameters are periodically updated by the student model with EMA, given by:

$$\theta_t = \alpha\theta_t + (1-\alpha)\theta_s$$

where $\alpha$ is a positive constant and is empirically fixed as 0.99 for Webpage/Wikigold and 0.995 for the remaining datasets. Finally, to train the entity pathway, we adopt a KL divergence loss on the student model's output logits and corresponding soft labels from the teacher model's prediction. The formulation is:

$$\mathcal{L}_{e\_cls}(\hat{\boldsymbol{y}}^s, f(\boldsymbol{x};\theta_s)) = \sum_i \sum_j -\hat{y}_{ij}^s \log f_{ij}(\boldsymbol{x};\theta_s) + f_{ij}(\boldsymbol{x};\theta_s)\log(f_{ij}(\boldsymbol{x};\theta_s))$$

Then, the student model's entity pathway is trained to approximate the soft labels. For the binary pathway, we calculate a standard binary cross-entropy loss, which is given by:

$$\mathcal{L}_{b\_cls}(\tilde{y}^b, \hat{y}^b) = -\tilde{y}^b \log \hat{y}^b - (1-\tilde{y}^b)\log(1-\hat{y}^b)$$

where $\tilde{y}^b$ is the given distant label and $\hat{y}^b$ is generated by student model's binary pathway.

## D Pseudo-Code of DesERT

We describe the overall training pipeline of DesERT in Algorithm 1.

**Algorithm 1** Training workflow of DesERT

---

**Input:** Training data $\mathcal{D} = \{(\boldsymbol{x}_i, \tilde{\boldsymbol{y}}_i)\}_{i=1}^M$ with distant labels; two sets of teacher-student networks, $\theta_{t1}, \theta_{s1}$ and $\theta_{t2}, \theta_{s2}$;

1:   $t \leftarrow 0$
     */* Selection and self-training */*
2: **while** $t < T_1$ **do**
3:     Get a batch $\mathcal{B} = \{(\boldsymbol{x}_i, \tilde{\boldsymbol{y}}_i)\}_{i=1}^B \subset \mathcal{D}$;
4:     **if** $t < k$ **then**
5:       $\mathcal{B}_b \cup \mathcal{B}_e \leftarrow \mathcal{B}$; //decoupled datasets
6:       $\overline{\mathcal{B}}_b^1 = \{\mathcal{B}_b^{l,1}, \mathcal{B}_b^{u,1}\} \leftarrow \text{Sel}(f(\theta_{t1}), \mathcal{B}_b)$;
7:       $\overline{\mathcal{B}}_e^1 = \{\mathcal{B}_e^{l,1}, \mathcal{B}_e^{u,1}\} \leftarrow \text{Sel}(f(\theta_{t1}), \mathcal{B}_b)$;
8:       $\overline{\mathcal{B}}_b^2 = \{\mathcal{B}_b^{l,2}, \mathcal{B}_b^{u,2}\} \leftarrow \text{Sel}(f(\theta_{t2}), \mathcal{B}_b)$;
9:       $\overline{\mathcal{B}}_e^2 = \{\mathcal{B}_e^{l,2}, \mathcal{B}_e^{u,2}\} \leftarrow \text{Sel}(f(\theta_{t2}), \mathcal{B}_b)$;
10:      Update $\theta_{s1}$ with $\{\overline{\mathcal{B}}_b^1, \overline{\mathcal{B}}_e^1\}$ by minimizing $\mathcal{L}$;
11:      Update $\theta_{s2}$ with $\{\overline{\mathcal{B}}_b^2, \overline{\mathcal{B}}_e^2\}$ by minimizing $\mathcal{L}$;
12:     **else**
13:       $\mathcal{X}_B \leftarrow \{(\boldsymbol{x}_i)\}_{i=1}^B$
14:       $\overline{\mathcal{B}}_b \cup \overline{\mathcal{B}}_e \leftarrow \text{Guess}(f(\theta_{t1}), f(\theta_{t2}), \mathcal{X}_B)$;
15:       Update $\theta_{s1}, \theta_{s2}$ with $\{\overline{\mathcal{B}}_b, \overline{\mathcal{B}}_e\}$ by minimizing $\mathcal{L}$
16:     **end if**
17:     $\theta_{t1} \leftarrow \text{EMA}(\alpha, \theta_{t1}), \theta_{t2} \leftarrow \text{EMA}(\alpha, \theta_{t2})$
18:     $t \leftarrow t + 1$
19: **end while**
     */* Post-hoc entity pathway finetuning */*
20: **while** $t < T_1 + T_2$ **do**
21:     Get a batch $\mathcal{B} = \{(\boldsymbol{x}_i, \tilde{\boldsymbol{y}}_i)\}_{i=1}^B \subset \mathcal{D}$;
22:     $\mathcal{X}_B \leftarrow \{(\boldsymbol{x}_i)\}_{i=1}^B, \tilde{\mathcal{Y}}_B \leftarrow \{(\tilde{\boldsymbol{y}}_i)\}_{i=1}^B$
23:     $\overline{\mathcal{B}}_e^1 \leftarrow \text{EntitySel}(f(\theta_{t1}), \mathcal{X}_B, \tilde{\mathcal{Y}}_B)$;
24:     $\overline{\mathcal{B}}_e^2 \leftarrow \text{EntitySel}(f(\theta_{t2}), \mathcal{X}_B, \tilde{\mathcal{Y}}_B)$;
25:     Finetuning $\theta_{s1}$ with $\overline{\mathcal{B}}_e^1$
26:     Finetuning $\theta_{s2}$ with $\overline{\mathcal{B}}_e^2$
27:     $\theta_{t1} \leftarrow \text{EMA}(\alpha, \theta_{t1}), \theta_{t2} \leftarrow \text{EMA}(\alpha, \theta_{t2})$
28:     $t \leftarrow t + 1$
29: **end while**

---

# E   Limitations

While DesERT has been proven to be effective for distant supervision, it is still subject to certain limitations. First, our WCE loss is estimated from pseudo-labels instead of the real ones. While we empirically find our WCE loss works well, its performance is theoretically restricted. One potential solution is to estimate a small validation set to remedy this problem, but we leave it as our future work. Second, while the imbalance between entity labels is located in Figure 1, our framework does not particularly integrate special components to explicitly overcome this problem. We believe it is not hard to draw inspiration from the recent achievement in long-tailed learning to further improve the NER performance. Lastly, since DesERT ensembles two sets of teacher-student networks as previous works did [9, 49], we should train peer-student models simultaneously and utilize the predictions from dual-teacher models iteratively, thus resulting in relatively higher training costs. We hope future efforts are made to alleviate the cost of network ensembling.

# F   Ethics Statement

While distant supervision is deemed a cheap way to collect and curate training data, the off-the-shelf and external knowledge bases steering the autonomous annotation procedure may include bias and

unfairness. Indeed, if one trains the model by these biased labels, it may unpleasantly yield unfair and biased predictions on the basis of characteristics like race, gender, disabilities, LGBTQ, or political orientation. Therefore, when deploying our DesERT framework, it is recommended to equip it with some auxiliary tools for labeling censorship so as to improve overall fairness and ethical standards. Grounded on this, we would suggest regarding our DesERT framework as an auxiliary weakly-supervised annotation tool for assisting human annotations.

