# OpenReview forum: "Debiased and Denoised Entity Recognition from Distant Supervision"
_NeurIPS.cc/2023/Conference — NeurIPS 2023 poster_

### Official Review · Reviewer_qqbH · 2023-06-15

**Soundness:** 2 fair
**Presentation:** 3 good
**Contribution:** 2 fair
**Rating:** 5
**Confidence:** 4

**Summary:**

The paper presents DesERT, a novel self-training framework for distant-supervised named-entity recognition (NER). It addresses the challenges of noisy distant labels and introduces two types of biases that impact performance. DesERT incorporates a dual-form self-training approach and a debiased module to improve sample selection and token representations. Experimental results demonstrate significant performance gains, surpassing existing methods on multiple benchmark datasets.




**Strengths:**

1. The problem of open-domain distantly supervised NER is important.

2. The paper is well-written.

3. The motivation of the paper is good.

**Weaknesses:**

1. Some important related works are missing, namely:

[1] Zhang, Wenkai, et al. "De-biasing Distantly Supervised Named Entity Recognition via Causal Intervention." Proceedings of the 59th Annual Meeting of the Association for Computational Linguistics and the 11th International Joint Conference on Natural Language Processing (Volume 1: Long Papers). 2021.

[2] Qu, Xiaoye, et al. "Distantly-Supervised Named Entity Recognition with Adaptive Teacher Learning and Fine-grained Student Ensemble." arXiv preprint arXiv:2212.06522 (2022).

[3] Jiang, Haoming, et al. "Named Entity Recognition with Small Strongly Labeled and Large Weakly Labeled Data." Annual Meeting of the Association for Computational Linguistics. 2021.

Specifically, [1] shares a very similar motivation with this work for *debiasing* distantly supervised NER. Please discuss the key difference between these baselines.

2. The method proposed in 4.3 is a direct application of [Chen et al., 2022], and the dual co-guessing framework is also a direct extension in [Liang et al., 2020] by adding two model branches. Overall, the authors need to justify the novelty of the proposed method.



**Questions:**

The experiments of ChatGPT are very interesting and I would recommend the authors to put it in the main content instead of appendix.

---

> ### Author Rebuttal · Authors · 2023-08-10
>
> Thanks very much for your valuable comments and suggestions! Our detailed responses are as follows.
>
> **Q1. The method proposed in 4.3 is a direct application of [Chen et al., 2022], and the dual co-guessing framework is also a direct extension in [Liang et al., 2020] by adding two model branches. Overall, the authors need to justify the novelty of the proposed method.**
>
> A. We would like to defend the novelty of our work since the purpose of our task-splitting mechanism is not to simply and blindly propose a new architectural approach.
>
> Rather, behind it, it manifests a significant issue of selection bias in the DSNER, that commonly exists but has not been uncovered before in the NER literature.
>
> Our approach is partially developed toward resolving this issue and delivered as a holistic DSNER system via decoupled learning and debiased training that establishes new SOTA performances.
>
> Indeed, we think that a simple and blindly-designed extension of existing techniques --- with no rigorous empirical findings and support --- wouldn’t have gotten us here.
>
> We sincerely hope that the reviewer could reassess our paper following the evidence chain from empirical insights on bias (Figure 1), holistic debiased DSNER framework (Section 4), to superior performance (Section 5).
>
> **Compared with [Chen et al., 2022]:**
>
> Technically, we made the first attempt to apply the WCE loss in the distant supervision regime where the labeled/unlabeled set can be dynamic. In contrast, the original paper [Chen et al., 2022] studies the conventional semi-supervised learning problem with fixed datasets.
>
> From the theoretical standpoint, we also provide new theoretical insights in our Appendix to show that WCE does indeed shrinks the variances, which is blank in the literature.
>
> **Compared with [Liang et al., 2020]**:
>
> Indeed, the high confidence selection is popular in weakly-supervised learning, e.g. FixMatch in semi-supervised learning. Our method extends it with an ensemble technique that further improves robustness. But, we have to say that this is not our core contribution. Instead, our work focuses on bringing new insights and findings on the bias problem in DSNER.
>
>
>
> **Q2. Some important related works are missing.**
>
> A. Thanks for your suggestions. We'll add the discussion on these related works in our revised version. We'd like to clarify that these works are quite different from ours.
>
> 1. First, [1] is fundamentally different from our work.
>
>     (i)-**Different Bias Issues**: [1] particularly studies two **structural causal biases** caused by the dictionaries, including *inter-dictionary-bias* that overlooks out-of-dictionary entities and *intra-dictionary bias* that different dictionaries can cause different model behaviors. In contrast, our paper studies the selection biases in the current selection&self-training schema.
>
>     (ii)-**Different Technical Solutions**: [1] further introduce a structural causal model (SCM) for DSNER that mainly Backdoor Adjustment and Causal Invariance Regularizer, both come from a causal perspective. Instead, our method still follows the sample selection and self-training framework.
>
>     (iii)-**Performance Gaps**: [1] largely underperforms our DesERT on empirical performance, e.g. 81.54 vs 86.95 (ours) F1 scores on CoNLL03.
>
> 2. We felt sorry for missing [2] because this AAAI'23 paper is officially published near the NeurIPS submission deadline. Technically, [2] and our paper shares similar selection and self-training frameworks. But, the core idea of our paper is rather different from [2] since our main novelty lies in the identification of two important biases in DSNER for the first time. Moreover, our DesERT algorithm enjoys better performance, e.g. 85.59 vs 86.95 (ours) F1 scores on CoNLL03 and 70.55 vs 72.73 (ours) F1 scores on Webpage.
>
> 3. [3] studies a different DSNER setup, where a large set of weak labels and a small set of clean training data. From a technical standpoint, [3] adopts a continual pretraining framework that jointly trains with clean and noisy entities.
>
> The different characteristics are summarized in the following table. We believe our work is quite different from these papers and brings new insights to the community.
>
> | Characteristics             | DesERT                           | [1]                     | [2]                      | [3]                    |
> | --------------------------- | -------------------------------- | ----------------------- | ------------------------ | ---------------------- |
> | Biased Problem              | Selection (Label) Biases | Causal Bias             | N/A                      | N/A                    |
> | Techniques                  | Selection&Self-Training          | Structural Causal Model | Selection&Self-Training  | Continual Pre-training |
> | Require Clean Training Data | No                               | No                      | No                       | Yes                    |
> | Performance                 | Best                             | Low                     | N/A (Different Datasets) | Good                   |
>
> **Q3. The experiments of ChatGPT are very interesting and I would recommend the authors to put it in the main content instead of appendix.**
>
> A. We are glad you enjoyed our new results on ChatGPT-based DSNER. In our revised version, we will move part of the discussions and add more supporting results to our main content.

---

> > ### Comment · Reviewer_qqbH · 2023-08-12
> > **Thanks for your rebuttal**
> >
> > I would like to thank the authors for the detailed response! I have updated my score accordingly.

---

> > > ### Author Response · Authors · 2023-08-12
> > > **Thank you!**
> > >
> > > Thank you very much! We will make sure to add the relevant discussion to the final version of this paper. : )

---

### Official Review · Reviewer_NjrY · 2023-07-07

**Soundness:** 3 good
**Presentation:** 3 good
**Contribution:** 2 fair
**Rating:** 5
**Confidence:** 3

**Summary:**

The paper proposes a novel self-training framework, DesERT, for distantly-supervised named entity recognition (NER) that addresses two types of biases omitted by prior work: structural noise in distant labels and the inherent bias introduced by the self-training framework. The paper shows that the proposed framework establishes a new state-of-the-art performance, with an improvement of +2.22% average F1 score on five standardized benchmarking datasets.

**Strengths:**

1.	The paper identifies the structural noise in the distant supervision for NER. Specifically, the imbalance lies (1) between the distribution of different types of entity and (2) between named entity and non-entity.
2.	The paper proposes a new framework for self-training when the pseudo label is noisy and sometimes unreliable.
3.	In experiments, the proposed method accomplishes new SOTA on five benchmarks of NER.


**Weaknesses:**

1.	The framework seems a simple mixture of existing techniques and their application on the NER task. The two observed omitted bias of previous works resembles the data bias the training bias proposed in [1]. The worse-case cross entropy loss also inherits [1] without much task-specific adaptation. The teacher network and its exponential moving average seem from [2]. I suggest the authors clearly claim where their novelty lies in this work.
2.	The writing is not very easy to follow. A high-level algorithm exhibiting the working flow of the proposed approach is suggested in the main document. In the current state, you just list all the techniques in each paragraph, and it is a little bit hard to get across the inner relationship. For example, there is a teacher network and a student network in Figure2 but the teacher network is not mentioned until implementation details.
3.	The double-head pathway requires an additional head and thus may cause extra inference latency. Analysis and discussion on this should be included thus we can see whether the inference speed declines obviously or not.
[1] Debiased Self-Training for Semi-Supervised Learning
[2]Improving Distantly-Supervised Named Entity Recognition with Self-Collaborative Denoising Learning


**Questions:**

1.	Is your backbone encoder (RoBERTa-base and DistilRoBERTa) consistent with the baseline method? Furthermore, are they comparable in parameter scale?
2.	I don’t understand the difference between the two peer networks in dual co-guessing mechanism. Are they only different with other in parameter initialization?


**Limitations:**

see weakness

---

> ### Author Rebuttal · Authors · 2023-08-10
>
> Thanks very much for your insightful comments and suggestions! Our detailed responses are as follows.
>
> **Q1. The framework seems a simple mixture of existing techniques and their application on the NER task. The two observed omitted bias of previous works resembles the data bias the training bias proposed in [1]. The worse-case cross entropy loss also inherits [1] without much task-specific adaptation. The teacher network and its exponential moving average seem from [2]. I suggest the authors clearly claim where their novelty lies in this work.**
>
>  A. We would like to defend the novelty of our work since the purpose of our task-splitting mechanism is not to simply and blindly propose a new architectural approach.
>
> Rather, behind it, it manifests a significant issue of selection bias in the DSNER, that commonly exists but has not been uncovered before in the NER literature.
>
> Our approach is partially developed toward resolving this issue and delivered as a holistic DSNER system via decoupled learning and debiased training that establishes new SOTA performances.
>
> Indeed, we think that a simple and blindly-designed extension of existing techniques --- with no rigorous empirical findings and support --- wouldn’t have gotten us here.
>
> We sincerely hope that the reviewer could reassess our paper following the evidence chain from empirical insights on the bias (Figure 1), holistic debiased DSNER framework (Section 4), to superior performance (Section 5).
>
> **Compared with [1] (WCE):**
>
> Technically, we made the first attempt to apply the WCE loss in the distant supervision regime where the labeled/unlabeled set can be dynamic. In contrast, the original paper [Chen et al., 2022] studies the conventional semi-supervised learning problem with fixed datasets.
>
> From the theoretical standpoint, we also provide new theoretical insights in our Appendix to show that WCE does indeed shrinks the variances, which is blank in the literature.
>
> **Claims on the teacher network technique [2]**:
>
> Sorry for confusing the reviewer. Actually, we didn't claim that the teacher-student network is our contribution since it originates from the SCDL framework. We simply put it in the Appendix to ensure the integrity of our work.
>
> **Q2. A high-level algorithm exhibiting the working flow of the proposed approach is suggested in the main document. There is a teacher network and a student network in Figure 2 but the teacher network is not mentioned until implementation details.**
>
>  A. Thanks for the suggestion. We've provided the pseudo-code of DesERT in Appendix D. In fact, our core components like decoupled training, debiased SSL, and co-guessing are general-purpose components and can be attached to any existing selection&self-training-based DSNER algorithms. To obtain SOTA performance, we chose the recent SOTA algorithm SCDL as our base implementation, which introduces the teacher-student network and some other tricks. However, we wouldn't like to claim such components as our core novelty and thus, only report it in our Appendix. We'll polish our manuscript in the final version.
>
> **Q3. The inference speed declines of double head.**
>
>  A. The double-head module is attached to the top of the backbone, and thus will not cause much extra inference latency. We conducted experiments to show the running time and the results are shown below,
>
> | Method      | Inference Testing Time (Seconds/Epoch) | Back Propagation Training Time (Seconds/Epoch) |
> | ----------- | -------------------------------------- | ---------------------------------------------- |
> | Single Head | 12.19                                  | 74.76                                          |
> | Double Head | 14.58                                  | 79.67                                          |
>
> It can be shown that our double-head pathway brings negligible extra computation costs.
>
> **Q4. Is your backbone encoder (RoBERTa-base and DistilRoBERTa) consistent with the baseline method? Furthermore, are they comparable in parameter scale?**
>
>  A. As we also stated in our responses to *Reviewer#WUVn*, our implementation and configurations directly follow SCDL [9] and BOND [10]. Therefore, our method has the same backbone, teacher-student architecture, and even mostly common parameters (the same learning rate, batch size, teacher updating coefficients, and so on). Therefore, the size of the parameters is also consistent with the baselines.
>
> **Q5. I don’t understand the difference between the two peer networks in dual co-guessing mechanism. Are they only different with other in parameter initialization?**
>
>  A. In the beginning, they have different initial weights. As the training proceeds, the two peer networks may tend to fit different noisy patterns from data, especially when the noise is not well rectified. Therefore, an ensemble of label guessing can bring new opportunities for filtering clean samples and improving their robustness.

---

> > ### Comment · Reviewer_NjrY · 2023-08-18
> > **Thanks for the rebuttal**
> >
> > I raise my score because some of my concerns were addressed in the rebuttal.

---

> ### Author Response · Authors · 2023-08-18
> **Sincerely look forward to further discussions**
>
> Dear reviewer, thanks for your comments and suggestions. We carefully addressed the concerns of all the reviewers in our rebuttal, and two of the other reviewers **qqbH** and **WYfp** have responded to the rebuttal and engaged in the discussion with us. We sincerely look forward to receiving your response. Thank you very much!

---

### Official Review · Reviewer_Nukd · 2023-07-08

**Soundness:** 3 good
**Presentation:** 3 good
**Contribution:** 2 fair
**Rating:** 5
**Confidence:** 4

**Summary:**

The paper studies the distantly-supervised named entity recognition (NER) problem and proposes a framework called DesERT. Motivated by the bias patterns in the distant labeling noise, DesERT conducts clean token selection to split the training set into a clean set and an unlabeled set and then trains the encoder model via debiased self-training with a worst-case cross-entropy (WCE) loss. The experiments are conducted on five standard NER benchmarks and DesERT demonstrates moderate improvements over baselines.

**Strengths:**

* Originality: The overall method design follows the common setup in distantly-supervised NER, by first selecting clean data and then conducting self-training, and is not novel. However, there are some details, including the analysis of the label noise distribution and the method for debiased self-training, that are new.
* Quality: The methods are well-motivated by the challenges in distantly-supervised NER and are reasonable designs.
* Clarity: The paper is clearly-written overall.
* Significance: The analyses and method proposed are specific to the distantly-supervised NER setting. In addition, the empirical performance of the method is not significantly better than the baselines. Therefore, the scope of the paper and its impact is quite narrowed to be considered for being published at NeurIPS. I feel that the paper could be more suitable for an NLP venue.

**Weaknesses:**

* Problem scope and method design: The distantly-supervised NER setting is a fine research setup, but if the proposed method is very specific and applicable only to such one setup (as in this paper), it still feels quite narrowed especially for an ML conference like NeurIPS. The impact and scope of the paper could be enriched if the method is applicable to more types of tasks where label noise is present (this will have to be shown via empirical results, not just conceptually).
* The empirical advantage of the proposed method over baselines is not very significant: In most cases, DesERT is only better than the previous best baseline by ~1 F1 point, with the most significant improvements on CoNLL03/Webpage. But it's possible that this is because the authors use either CoNLL03 or Webpage for their method development.
* Comparisons with LLMs: I understand that LLMs are not typically used in IE tasks since they are not very parameter-efficient for them. However, given the current rapid development in LLMs, it'd have been nice to include some discussions and empirical results of using LLMs for NER tasks. I noticed that some results are in the Appendix, but are not in the main paper.

**Questions:**

RoSTER is discussed in the related work but not included in the baseline comparisons. Is there a particular reason for that?

**Limitations:**

The authors discussed limitations in the Appendix.

---

> ### Author Rebuttal · Authors · 2023-08-10
>
> Thanks very much for your comments and suggestions. Below we address the feedback and comments in detail:
>
> **Q1. Problem scope and method design.**
>
> A. We originally tend to focus on the setting of distantly-supervised NER, which is a practical and fundamental task in NLP and benefits many downstream applications (e.g., entity linking and relation extraction). But, the core debiasing components of DesERT can be generalized to more sequential prediction tasks, even beyond NLP tasks (e.g. action recognition for video, speech recognition).
>
> To see this, we closely follow [C1] on the sequence labeling benchmarks SNIPS and ATIS in a quest to make the comparison fair. We followed the literature on noisy label learning and injected 20%/30% synthetic noise. The results in F1 are shown below. It says that DesERT performs way better than its counterparts, effectively proving its superiority in handling such weakly-supervised sequential tagging tasks.
>
> | Methods | SNIPS (20% noise) | SNIPS (30% noise) | ATIS (20% noise) | ATIS (30% noise) |
> | ------- | --------------- | --------------- | -------- | -------- |
> | DesERT  | 94.29           | 93.51           | 95.15    | 94.73    |
> | RoBERTa | 85.03           | 77.09           | 90.89    | 90.05    |
>
> Moreover, We conducted a simple experiment on the bioinformatics tasks for predicting the second structures of the Proteins and we refer the reviewer for our responses to *Reviewer#WYfp* for details.
>
> We hope these new results can alleviate the reviewer's concern about the ability of DesERT for a broader context of problem setups. And we'll incorporate all-the-above into our next revision.
>
> Further, we want to provide a further defense to our paper.  As we dive into the literature, despite the criticism from the reviewer, there has been a number of papers --- dedicated to the DSNER problem --- that manage to be published in top-tiered AI conferences. Posited by many of these papers, DSNER often acts as an iconic, profound, and representative task for the track of noisy-labeled NLP which extends to a variety of sequence labeling tasks.
>
> *[C1] Li, et al. Handling rare entities for neural sequence labeling[C]//ACL. 2020: 6441-6451.*
>
> **Q2. The empirical advantage of the proposed method.**
>
> A. To recap a bit, despite what was mentioned by the reviewer, our approach achieves quite notable performance gain on CoNLL03 (+3.26 F1) and Webpage (+4.26 F1).
>
> We further want to refer the reviewer to our response to Q1, where our approach performs significantly better than the rivals, on another sequence labeling task (we did this experiment this week as per the request from the reviewers, so we didn’t perform any dense tuning ;) )
>
> At last, despite that our approach did achieve SOTA performances on various benchmarks of DSNER, perhaps we humbly ask the reviewer not to attribute heavily to the empirical numbers of a method. This is because wholeheartedly, we believe that a method up to SOTA is not the sole aspect of a NeurIPS submission.
>
> **Q3. Comparisons with LLMs.**
>
>  In our original submission, we provided a thorough discussion on how recent LLMs can help resolve the DSNER problem without human supervision.
>
> Below, we summarize some takeaways:
>
> 1. **The zero-shot performance of ChatGPT is far from resolving the NER task.** As shown in Table 3 (Appendix), zero-shot ChatGPT exhibits 66.47 F1-score on  CoNLL03, which legs even behind the conventional distantly-supervised NER methods like our proposed DesERT. Notably, this is also supported by recent works [C2], which find that ChatGPT significantly underperformed the fine-tuned small language models in many NLP applications.
> 2. **Self-generated demonstration can improve the ChatGPT on NER.** The performance of ChatGPT on unsupervised NER can be alleviated by an in-context learning algorithm. Specifically, we generate demonstration sentences by ChatGPT-given entity labels. Such a strategy can improve the F1-score to 70.22 without any human supervision.
> 3. **ChatGPT prediction can serve as a new source of distant supervision.** Though the ChatGPT prediction is not satisfactory for practical employment, we find it is a good source of distant supervision for distilling a small language model. As shown in Table 3 (Appendix), DesERT obtains 79.53 F1 on such ChatGPT distant labels. Moreover, we also tested a hybrid supervision paradigm that mixes ChatGPT supervision and conventional knowledge-base supervision. We find that DesERT is able to achieve better performance (88.08 F1) than merely using knowledge base supervision (86.75 F1).
>
> In our response PDF submission, we expound upon our findings by presenting further experimental outcomes from the Wikigold dataset, which substantiate our assertions. As a result, **we firmly believe that DSNER remains a pivotal research pursuit in the LLM era**. Looking forward, we envision LLMs serving as a novel source of distant supervision, facilitating the distillation of distantly-supervised SLMs, which has enhanced accuracy and computational efficiency. We will move part of the discussions and add more supporting results to our main content.
>
> *[C2] Yejin Bang, et al. A multitask, multilingual, multimodal evaluation of chatgpt on reasoning, hallucination, and interaeictivity. CoRR, abs/2302.04023, 2023.225*
>
> **Q4. RoSTER is discussed in the related work but not included in the baseline comparisons. Is there a particular reason for that?**
>
> A. We apologize for this confusion. We closely followed the implementation and configurations of SCDL and BOND. Thus, all the baseline performances are from the corresponding papers (SCDL's baseline results copied BOND). In RoSTER, a different tagging scheme of the IO format (i.e., only distinguishing whether a token is part of an entity or not) is employed instead of the BIO scheme adopted by the other baselines. As a result, RoSTER is not included in the baseline comparisons to ensure the comparisons are fair.

---

> ### Author Response · Authors · 2023-08-18
> **Sincerely look forward to further discussions**
>
> Dear reviewer, thanks again for providing valuable suggestions. Two of the other reviewers **qqbH** and **WYfp** have responded to the rebuttal and engaged in the discussion with us. We haven’t heard back from you. Could you let us know? Thanks.

---

> > ### Comment · Reviewer_Nukd · 2023-08-19
> >
> > I appreciate the authors' response and the additionally provided results. I believe that they enhance the contribution and potential applicability to a wider range of tasks. Hence, I'm updating my rating in light of this.

---

> > > ### Author Response · Authors · 2023-08-19
> > >
> > > Thanks for your reply! We also sincerely thank you for your valuable time on our paper and thank you for supporting our paper.

---

### Official Review · Reviewer_WYfp · 2023-07-09

**Soundness:** 3 good
**Presentation:** 3 good
**Contribution:** 3 good
**Rating:** 6
**Confidence:** 5

**Summary:**

This paper presents a new method named DesERT for distantly supervised named entity recognition (NER), which starts with analyzing the noise of and biases in distant supervision which is used in self-training frameworks: 1) skewed data distribution, 2) inherent error patterns in pseudo-labels. They decompose the original NER task to a task-data taxonomy, with each component from this taxonomy manifesting an individual training paradigm. On several NER datasets, the proposed method outperforms many previous baseline methods.

**Strengths:**

- The method is well motivated based on solid analysis in self-training and distant supervision.
- The evaluation is done on five common NER datasets and they show great performance on all of them, better than recent existing methods.


**Weaknesses:**

- The presentation can be further improved with more illustrative examples and figures. The current presentation does not clearly introduce the biases and how the proposed method works.
- The paper lacks a discussion about how recent LLMs could solve the (distantly supervised) NER problem. With the advanced zero/few-shot generalization ability of modern LLMs, do you still even need such a complex framework for NER? If it is still needed, in which extent? These are quite important messages to the audience, but none of them are covered. I saw some content in appendix about ChatGPT, but not sure about the concrete conclusions that you want to tell the readers.
- The paper is rather limited to a special setting and task: distant supervised NER. To gain more interests of NeurIPS readers, I would suggest the authors generalize the paper to a broader context: sequence tagging, and do more experiments in bioinformatics tasks, considering DNA as sequences.

**Questions:**

- See weakness.

**Limitations:**

In the appendix.

---

> ### Author Rebuttal · Authors · 2023-08-10
>
> Thanks very much for your insightful comments and suggestions! Our detailed responses are as follows.
>
> **Q1. The presentation can be further improved with more illustrative examples and figures.**
>
> A. Thanks for your suggestions! We provided our visualized illustration of Dataset Taxonomy in Appendix A and the pseudo-code in Appendix D. We'll carefully revise our manuscript and provide more illustrative examples and figures in our revised version!
>
> **Q2. With the advanced zero/few-shot generalization ability of modern LLMs, do you still even need such a complex framework for NER? If it is still needed, in which extent?**
>
> A. In our original submission, we provided a thorough discussion on how recent LLMs can help resolve the DSNER problem without human supervision. Below, we summarize some takeaways:
>
> 1. **The zero-shot performance of ChatGPT is far from resolving the NER task.** As shown in Table 3 (Appendix), zero-shot ChatGPT exhibits 66.47 F1-score on  CoNLL03, which legs even behind the conventional distantly-supervised NER methods like our proposed DesERT. Notably, this is also supported by recent works [C1], which find that ChatGPT significantly underperformed the fine-tuned small language models in many NLP applications.
> 2. **Self-generated demonstration can improve the ChatGPT on NER.** The performance of ChatGPT on unsupervised NER can be alleviated by an in-context learning algorithm. Specifically, we generate demonstration sentences by ChatGPT-given entity labels. Such a strategy can improve the F1-score to 70.22 without any human supervision.
> 3. **ChatGPT prediction can serve as a new source of distant supervision.** Though the ChatGPT prediction is not satisfactory for practical employment, we find it is a good source of distant supervision for distilling a small language model. As shown in Table 3 (Appendix), DesERT obtains 79.53 F1 on such ChatGPT distant labels. Moreover, we also tested a hybrid supervision paradigm that mixes ChatGPT supervision and conventional knowledge-base supervision. We find that DesERT is able to achieve better performance (88.08 F1) than merely using knowledge base supervision (86.75 F1).
>
> In our response PDF submission, we expound upon our findings by presenting further experimental outcomes from the Wikigold dataset, which substantiate our assertions. As a result, **we firmly believe that DSNER remains a pivotal research pursuit in the LLM era**. Looking forward, we envision LLMs serving as a novel source of distant supervision, facilitating the distillation of distantly-supervised SLMs, which has enhanced accuracy and computational efficiency. We will move part of the discussions and add more supporting results to our main content.
>
> *[C1] Yejin Bang, et al. A multitask, multilingual, multimodal evaluation of chatgpt on reasoning, hallucination, and interaeictivity. CoRR, abs/2302.04023, 2023*
>
> **Q4. Generalize the paper to a broader context: sequence tagging, and do more experiments in bioinformatics tasks.**
>
> A. We originally tend to focus on the setting of distantly-supervised NER, which is a practical and fundamental task in NLP and benefits many downstream applications (e.g., entity linking and relation extraction). But, the core debiasing components of DesERT can be generalized to more sequential prediction tasks, even beyond NLP tasks (e.g. action recognition for video, speech recognition).
>
> To see this, we closely follow [C2] on the sequence labeling benchmarks SNIPS and ATIS in a quest to make the comparison fair. We followed the literature on noisy label learning and injected 20%/30% synthetic noise. The results in F1 are shown below. It says that DesERT performs way better than its counterparts, effectively proving its superiority in handling such weakly-supervised sequential tagging tasks.
>
> | Methods | SNIPS (20% noise) | SNIPS (30% noise) | ATIS (20% noise) | ATIS (30% noise) |
> | ------- | --------- | --------- | -------- | -------- |
> | DesERT  | 94.29  | 93.51| 95.15| 94.73 |
> | RoBERTa | 85.03 | 77.09| 90.89| 90.05  |
>
> To concur with the request made by the reviewer, despite within the short period of time of this week to prepare the rebuttal and lacking of expertise, we supplement another set of the experiments from another domain --- bioinformatics --- that is completely unrelated to the NER. We conducted a simple experiment on the bioinformatics dataset FLIP that predicts the second structures of the proteins [C3]. We injected 20% noise into synthetic datasets. Owing to the time limits, we adapted the decoupled training and debiased SSL modules to the benchmark from ProtTrans. In particular, we use a 30-layered ProtBERT model as the backbone and only fine-tuned the top 5 layers due to limited computation sources. We found that the ProtBERT is quite robust since we only fine-tune 5 layers. But, DesERT still shows some benefits to improve the robustness.
>
> | Methods               | FLIP dataset (Valid Acc) |
> | --------------------- | ------------------------ |
> | ProtBERT (Supervised) | 80.83|
> | ProtBERT (20% noise)  | 69.03|
> | DesERT (20% noise)  | 70.38|
>
> We hope these new results can alleviate the reviewer's concern about the ability of DesERT for a broader context of problem setups. We'll incorporate it into our revision.
>
> Further, we want to provide a further defense to our paper.  As we dive into the literature, despite the criticism from the reviewer, there has been a number of papers --- dedicated to the DSNER problem --- that manage to be published in top-tiered AI conferences. Posited by many of these papers, DSNER often acts as an iconic, profound, and representative task for the track of noisy-labeled NLP which extends to a variety of sequence labeling tasks.
>
> *[C2] Li, et al. Handling rare entities for neural sequence labeling[C]//ACL. 2020: 6441-6451.*
>
> *[C3] Ahmed, et al. ProtTrans: Towards Cracking the Language of Lifes Code Through Self-Supervised Deep Learning and High Performance Computing[J]//TPAMI. 2021*

---

> > ### Comment · Reviewer_WYfp · 2023-08-17
> >
> > Thank you for your new additional experiments and explanation. I have raised my score.

---

> > > ### Author Response · Authors · 2023-08-18
> > >
> > > Thanks for your valuable suggestions that make our paper more solid. We will incorporate the new results and the fruitful points in our future revision.

---

### Official Review · Reviewer_WUVn · 2023-07-09

**Soundness:** 3 good
**Presentation:** 4 excellent
**Contribution:** 3 good
**Rating:** 7
**Confidence:** 3

**Summary:**

The paper introduced DesERT, a new method that improves the distant supervision training framework for DSNER (distant supervision NER) using two highly effective components: 1. To solve the skewed label distribution (entity vs. non-entity tokens) in DSNER, DesERT proposes to decompose learning into two parts: a `binary` setup where the model learns to distinguish entities from non entities, and an `inter-entity` setup where the task is to predict the specific entity category. 2. To improve pseudo-label quality and alleviate the harmful effects of confirmation bias, DesERT proposes to use a worst-case cross-entropy (WCE) loss during training. The proposed approach is tested thoroughly on 5 datasets where it obtains consistent improvements over baselines.

**Strengths:**

- The presentation of the paper is clear and concise. The method is easy to understand and the delivery of the paper is smooth.

- The approach is extremely well-motivated; each proposed component has detailed reasoning into why this component was chosen and what problem it solves. For instance, through analysis, the authors state that true non-entity tokens are (unsurprisingly) far more frequent than entity tokens. Due to this imbalance, they notice that these non-entity tokens are picked up much more frequently and that entity tokens (which are the main focus of NER) are largely overlooked. To address this drawback, they propose to split the task in two subtasks: `entity vs. non-entity prediction` and `inter-entity prediction`, which is intuitive and well-motivated.

- The experiments are comprehensive. The method is compared with numerous baselines and 5 datasets.

- The analyses of the method are thorough as well. The authors analyze the benefits of each component through an ablation study and also provide theoretical insights into the WCE loss.

**Weaknesses:**

- The results come without a sense of variance. Are the performance improvements significant? What is the standard deviation of your approach compared to the baselines?

- The paper presents detailed information about the hyperparameters used. It seems that the values of these parameters are quite specific. For instance, the co-guessing is performed starting from epochs 6,40,35,30,30 on the five datasets.  Can you elaborate more on how all these hyperparameters were chosen and the search space used? My main concern is that DesERT is extremely thoroughly tuned while the baselines are not. Can you detail the tuning of the baselines as well?

The previous concerns make me uncertain if the reported improvements come from excessive tuning or if they are significant at all. I am willing to adjust my score if this concern/these questions are addressed.

**Questions:**

Please see the questions from the `Weaknesses` section. Additionally:

- You mentioned that splitting into clean/noisy is done on the fly. I would be curious to see how frequently it happens that at a later iteration a token that was clean is not clean anymore.

- It seems that post-hoc tuning is not that effective  on most datasets and it may be significant only on Wikigold? Do you have some insights on this? What’s special about Wikigold?

- You mentioned that only 2 epochs are used for post-hoc tuning. How was this value chosen? Does the performance degrade if more training is performed here?

**Limitations:**

I do not see a specific discussion of limitations. I suggest adding such a discussion. Additionally, I don't see any potential negative societal impact of this work.

---

> ### Author Rebuttal · Authors · 2023-08-10
>
> Thank you very much for your comments and suggestions! We are happy you enjoyed the paper. Below are our detailed responses:
>
> **Q1. The results come without a sense of variance. Are the performance improvements significant? What is the standard deviation of your approach compared to the baselines?**
>
> A.  Sorry for missing the variance. Our implementation and configurations closely adhere to the codes of SCDL [9] and BOND [10]. Consequently, our baseline results align with those presented in the respective source papers. In order to maintain uniformity in formatting, we have refrained from reporting the variance, due to the absence of such information in the cited references. Below, we present the standard deviation values corresponding to our initial results:
>
> | Method          | CoNLL03 | OntoNotes5.0 | Webpage | Wikigold | Twitter |
> | --------------- | :-----: | :----------: | :-----: | :------: | :-----: |
> | DesERT          |  0.22   |     0.43     |  0.40   |   0.46   |  0.32   |
> | DesERT-Finetune |  0.17   |     0.20     |  0.22   |   0.30   |  0.17   |
>
> It can be shown that our variances are not substantial. In view of the notable performance gap (average +2.22 F1), we believe that our algorithm has shown statistically significant improvement. We'll add this to our revision.
>
>
>
> **Q2. The paper presents detailed information about the hyperparameters used. It seems that the values of these parameters are quite specific. For instance, the co-guessing is performed starting from epochs 6,40,35,30,30 on the five datasets. Can you elaborate more on how all these hyperparameters were chosen and the search space used? My main concern is that DesERT is extremely thoroughly tuned while the baselines are not. Can you detail the tuning of the baselines as well?**
>
>  A. Actually, we didn't spend much time tuning our DesERT since it is based on SCDL. Notably, our novel components only bring three extra parameters $\tau$, the number of fine-tuning epochs, and the start epoch for co-guessing $k$, which is tuned via a validation set. Other parameters, including learning rate, batch size, and teacher network updating parameters all follow SCDL, i.e. these seemingly 'specific values'. We believe these parameters have been well-tuned by the authors of SCDL [9] and thus, didn't modify them.
>
> We have to acknowledge that our approach to selecting the co-guessing epoch could be perceived as somewhat arbitrary, lending an unconventional aspect to these values. For CoNLL03, we tested $k=6$ and find the performance has been good enough. For other datasets, we found that it is proper to tune it by $k\in\\{30,35,40\\}$. Below we show the ablation on $k$ (feel free to reproduce these results by using our source code). It can be observed that DesERT is not very sensitive to $k$.
>
> | $k$  | CoNLL03 | Twitter |
> | :--: | :-----: | :-----: |
> |  15  |  86.72  |  52.22  |
> |  20  |  86.52  |  52.15  |
> |  25  |  86.78  |  52.21  |
> |  30  |  86.82  |  52.10  |
> |  35  |  86.48  |  52.44  |
> |  40  |  86.18  |  51.54  |
>
>
>
> **Q3. The previous concerns make me uncertain if the reported improvements come from excessive tuning or if they are significant at all. I am willing to adjust my score if this concern/these questions are addressed.**
>
> A. We hope the previous discussion has adequately addressed your concerns. In fact, our core components like decoupled training and debiased SSL are general-purpose components for sequential prediction tasks; see our responses to *Reviewers#WYfp/Nukd*. However, it has been an unspoken rule in the community that one paper has to be SOTA to be accepted (while not NeurIPS's policy). Therefore, we chose the recent SOTA algorithm SCDL as our base implementation. With guaranteed fair comparisons, we believe we've shown the benefit of our framework in improving DSNER.
>
>
>
> **Q5. You mentioned that splitting into clean/noisy is done on the fly. I would be curious to see how frequently it happens that at a later iteration a token that was clean is not clean anymore.**
>
>  A.  We found that only a small proportion of clean samples will be moved to the noisy set at later iterations. We may refer the reviewer for our submitted response PDF for the plotted confusion matrices across the epochs.
>
>
>
> **Q6. It seems that post-hoc tuning is not that effective on most datasets and it may be significant only on Wikigold? Do you have some insights on this? What’s special about Wikigold?**
>
>  A. We found that the reason is that Wikigold contains more false entity tokens than other datasets and thus the filtering ability of binary head helps it improve a lot. Particularly, Wikigold has 697 false entity tokens out of 2975 distantly labeled entities, whereas CoNLL03 records a count of 492/23649 in the same regard.
>
>
>
> **Q7. You mentioned that only 2 epochs are used for post-hoc tuning. How was this value chosen? Does the performance degrade if more training is performed here?**
>
>  A. This is tuned via the validation set. Yes, we find that too many epochs for fine-tuning may result in slight performance degradation on some datasets, which we suppose may be caused by the overfitting of excessive training.
>
>
>
> **Q8. I do not see a specific discussion of limitations. I suggest adding such a discussion. Additionally, I don't see any potential negative societal impact of this work.**
>
> A.   We provided our limitations and societal impact sections in Appendix E&F. 😊

---

> > ### Author Response · Authors · 2023-08-19
> > **Discussion**
> >
> > Dear Reviewer,
> >
> > Hi! As we are quite near the end of the discussion phase, we hope to make a slight nudge in pursuit of a response from you :)
> >
> > Luckily it seems all of the reviewers have reached a consensus of positivity towards our paper.
> >
> > We wonder if the reviewer has had a chance to check on our response. If have you any further questions or concerns, please let us know while we can still help clarify.
> >
> > Thank you!

---

> ### Comment · Area_Chair_reMg · 2023-08-20
> **Reviewer please acknowledge having read the rebuttal**
>
> Review WUVn please let me know if your questions are addressed and need to revise your score. Also, the authors have attached a pdf with additional results in response to your questions. Please check it here: https://openreview.net/forum?id=FAGY52HbyV&noteId=D39yj3aWgs

---

> > ### Comment · Reviewer_WUVn · 2023-08-20
> >
> > Thanks for spending the time to create such a strong rebuttal! The response addressed my initial concerns. After also reading the other responses/comments, I believe this is a strong piece of work and I raise my score accordingly.

---

> > > ### Author Response · Authors · 2023-08-20
> > >
> > > Thank you so much for your reply! We appreciate your time for providing insightful comments, which definitely help us a lot for improving our work.

---

### Author Rebuttal · Authors · 2023-08-10

We sincerely appreciate all reviewers for their great efforts in providing constructive and valuable feedback. We are pleased that the reviewers find our work **well-written** (**R1,R3**), **theoretically solid** (**R1,R2**), and **experimentally comprehensive** (**R1,R2,R4,R5**). Moreover, we are more than encouraged that all the reviewers agree our work is **well motivated that brings some new findings on biases to improve the DSNER problem** (**R1,R2,R3,R5**).

We are also happy that our experiments (in the Appendix) on recent LLMs like ChatGPT raises the interest of the reviewers (**R2,R3,R5**).

We have addressed the reviewers’ comments and concerns in individual responses to each reviewer. We also submitted a response PDF that shows:

[1] Figure 1: (R1Q5) Confusion matrices of data split across the epochs

[2] Figure 2: (R1Q6) Confusion matrix for Wikigold

[3] Table 1: (R2Q2&R3Q3) Additional Results with ChatGPT Supervision

(As abbreviations, we refer to reviewers **WUVn** as R1, **WYfp** as R2, **Nukd** as R3, **NjrY** as R4 and **qqbH** as R5 respectively.)

---

### Author Response · Authors · 2023-08-17
**Please let us know if you have any further questions regarding our rebuttal**

Dear reviewers,

We thank the chairs and reviewers for their hard work and constructive suggestion. As we move closer to the end of the discussion panel, we haven't heard back from the reviewers. Please let us know if you have any further questions regarding our rebuttal.

Sincerely,

Authors

---

### Decision · Program_Chairs · 2023-09-21

**Decision:**

Accept (poster)

**Comment:**

The paper introduces a novel self-training framework called DesERT to address the challenges of distant supervision in NLP tasks, particularly named-entity recognition (NER). It identifies two types of biases in distant labels that previous work overlooked, leading to suboptimal performance in distant-supervised NER. DesERT adapts the sample-selection process to the structural bias in distant labels and incorporates a debiased module to enhance token representations, improving the quality of pseudo-labels. Extensive experiments demonstrate that DesERT achieves a significant +2.22 average F1 score improvement on five benchmark datasets, establishing a new state-of-the-art performance in distant-supervised NER. Reviewers generally find the work well-written, well motivated, and with sound experiments; and thus support the acceptance of the paper.